# Stretchable piezoelectric biocrystal thin films

Jun Li[1], Corey Carlos[1], Hao Zhou[2], Jiajie Sui[1], Yikai Wang[1], Zulmari Silva-Pedraza[1,3], Fan Yang [4], Yutao Dong[1], Ziyi Zhang [1], Timothy A. Hacker[5], Bo Liu[3], Yanchao Mao [2] ✉ & Xudong Wang [1] ✉

Stretchability is an essential property for wearable devices to match varying strains when interfacing with soft tissues or organs. While piezoelectricity has broad application potentials as tactile sensors, artificial skins, or nanogenerators, enabling tissue-comparable stretchability is a main roadblock due to the intrinsic rigidity and hardness of the crystalline phase. Here, an amino acid-based piezoelectric biocrystal thin film that offers tissue-compatible omnidirectional stretchability with unimpaired piezoelectricity is reported. The stretchability was enabled by a truss-like microstructure that was self-assembled under controlled molecule-solvent interaction and interface tension. Through the open and close of truss meshes, this large scale biocrystal microstructure was able to endure up to 40% tensile strain along different directions while retained both structural integrity and piezoelectric performance. Built on this structure, a tissue-compatible stretchable piezoelectric nanogenerator was developed, which could conform to various tissue surfaces, and exhibited stable functions under multidimensional large strains. In this work, we presented a promising solution that integrates piezoelectricity, stretchability and biocompatibility in one material system, a critical step toward tissue-compatible biomedical devices.

Piezoelectricity is a common biophysical property that can be found in many physiological processes in living beings[1–3]. It couples strain with electrical polarization in biological systems to achieve many important functions, such as sensing, communication, and modulation. One representative example is the Piezo 1 and Piezo 2 ion channels that manipulate mechanotransduction of cells by using conformational changes under mechanical stimuli to control ion permeability and selectivity[4]. Therefore, introducing the piezoelectric effect in biological systems has been considered as a promising approach to the creation of human-machine interfaces that mimic or intervene the physiological

processes, and eventually enable effective diagnostics and therapeutics[5,6]. Nevertheless, biological applications of the piezoelectric effect still face many challenges, even though the study of piezoelectric materials has been active for over a century. One unmet goal is the creation of piezoelectric biomaterials with tissue-comparable stretchability[7]. Although stretchable piezoelectric devices based on ceramic composite and synthetic polymers have been reported[8–10], the environmentally unsustainable components, lack of desired biocompatibility and unmatched mechanical properties with biological systems prevents their practical applications in interfacing bio-tissues.

[1]Department of Materials Science and Engineering, University of Wisconsin-Madison, Madison, WI 53706, USA. [2]Key Laboratory of Materials Physics of Ministry of Education, School of Physics and Microelectronics, Zhengzhou University, Zhengzhou 450001, China. [3]Department of Surgery, School of Medicine and Public Health, University of Wisconsin-Madison, Madison, WI 53705, USA. [4]Department of Orthopaedics, Shanghai Key Laboratory for Prevention and Treatment of Bone and Joint Diseases, Shanghai Institute of Traumatology and Orthopaedics, Ruijin Hospital, Shanghai Jiao Tong University School of Medicine, Shanghai, Shanghai 200025, China. [5]Cardiovascular Research Center, University of Wisconsin–Madison, Madison, WI 53705, USA. ✉e-mail: ymao@zzu.edu.cn; xudong.wang@wisc.edu

Because tissues and organs are soft, and constantly moving or deforming (e.g., 20–30% deformation for skin[11]), omnidirectional stretchability is a critical property needed by all biomedical devices interfacing with the human body[12–15]. Driven by this goal, tremendous efforts have been devoted to enable intrinsic or extrinsic stretchability in conductive and semiconducting materials through structure innovation[16,17], molecular design[18,19], or interface engineering[20,21]. Initial successes in this field have already brought a decade booming of soft and stretchable bioelectronics in a large variety of applications covering neuron modulation, physiological sensing, disease diagnoses and treatments[22,23]. However, enabling the tissue-like stretchability is rather challenging in piezoelectric materials. As the piezoelectricity is a result of long-range ordering of internal molecular or ionic dipoles, a complete crystalline phase with aligned polarization is necessary to achieve the desired performance, which intrinsically and inevitably makes the material rigid, fragile, with a rather weak strain tolerance[24–26]. This challenge also persists in the emerging piezoelectric biocrystals, such as amino acids, peptides, and cellulose, though their excellent biocompatibility and biodegradability make them ideal choices for implantable devices[27–31]. Unlike rigid inorganic crystals, such as silicon (fracture limit <1%[32]) that can go through complex microfabrication procedures to create micro-patterns to enable desired stretchability, piezoelectric biocrystals are incompatible with those energy-intensive processes due to their relatively weak intermolecular bonding and temperature-sensitive chemistry[31,33,34]. How to enable tissue-compatible stretchability while keeping their structure and property intact is a key roadblock for piezoelectric biocrystals to reach their promising future of practical applications.

Here, we report a large-scale omnidirectionally stretchable piezoelectric biocrystal network self-assembled from DL-alanine amino acid. Different from synthetic piezoelectric biomaterials obtained by complex chemical reactions (e.g., Poly-l-lactic Acid (PLLA)[35,36]), DL-alanine is a naturally occurring substance with excellent abundance and sustainability and is involved in metabolism of many biological processes[37,38]. It possesses a superior piezoelectric response among akin biomaterials[39,40]. DL alanine is one of the strongest piezoelectric amino acids. Unlike the D or L alanine that only have weak shear piezoelectricity, DL alanine crystalizes with its molecular dipoles well aligned in both longitudinal and transverse directions, leading to high piezoelectric coefficients. In our biocrystal DL-alanine thin film, the tissue-like stretchability is enabled by a continuous truss-like pattern formed by interconnected DL-alanine microfibers (MF) via supramolecular packing. The open-mesh structure of the continuous and ordered crystalline network enables unprecedented omnidirectional stretchability that is orders of magnitude higher than the fracture limit of bulk crystals; while still preserves a spontaneous and uniform piezoelectric polarization over the entire film, distinguishing from networks formed by microfibers which can hardly offer such continuity to sustain a good piezoelectricity when stretched. This tissue-comparable stretchability and stable piezoelectricity enable the development of stretchable, implantable and degradable electromechanical devices that can conform to the irregular tissue surfaces and stably function under large strain situations. Compared to other works on conformal piezoelectric devices (Table S1)[8–10,41–44], our major advantages are omnidirectional stretchability and conformability together with easy synthesis based on self-assembly.

## Results

### DL-alanine crystalline network synthesis and growth mechanism

The truss-like pattern of interconnected DL-alanine MFs was self-assembled on a hydrophilic substrate when it was slowly lifted up from a DL-alanine water-ethanol biphasic solution with a certain mixing ratio (Fig. S1, experimental details are included in Methods). The as-received film exhibited a uniform and interconnected MF network across the entire substrate surface over centimeters and beyond (Fig. 1a). All the

MFs were partially aligned along the pulling direction, with a repeating bifurcating and joining feature that resulted in an interconnected truss-like network. In this microstructure, DL-alanine MFs formed triangular or quadrilateral units with an average open mesh size of 100–500 μm in length and 20-100 μm in width. A typical DL-alanine MF exhibited a trapezoid-shaped belt morphology that was 2-3 μm wide and ~400 nm thick (Fig. 1b, Top). A fibrous fine feature could be observed from the top surfaces, indicating the MFs were assembled by nanofibril building blocks (Fig. 1b, bottom). As the unique growth behavior that led to the truss-like structure, the MFs bifurcated or merged smoothly without showing any overlaying morphology, or significant change to their sizes (Fig. 1c). Atomic force microscopy (AFM) topography further revealed a clear and sharp interface at the intersections following the side wall of one MF, but no gap could be detected there (Fig. 1d). Consecutive merging and bifurcation could also lead to an X-shaped junction (Fig. S2). The angles between the two joining/branching MFs were measured in the range of 15°-20°, consistent across the entire network. This small contact angle led to a large interfacial area at the junctions, which is favorable for achieving high structural stability. The bifurcation and remerging of MFs occurred repeatedly and constantly during the growth process, yielding a uniformly distributed truss-like pattern across the entire substrate area.

The crystallinity of the DL-alanine MF network was examined by X-ray diffraction (XRD). Different from powder samples, the truss-like DL-alanine MF network only exhibited three strong diffraction peaks at 16.40°, 20.79°, and 33.22°, corresponding to the (110), (210) and (410) planes of the piezoelectric Pna2$_1$ phase (Figs. 1e and S3), respectively, indicating a well-crystallized structure with likely aligned orientations. The crystallographic alignment of the MF network was further validated by the two-dimensional (2D) XRD (Fig. 1f). The 2D XRD image reveals more detailed information about the crystal structure, such as orientation and texture. Crystal orientation can be indicated by the pattern of the ring. The network exhibited a concentrated intensity at 20.79°, suggesting that majority of the MFs were aligned along the same direction, whereas the powder samples showed a homogeneous diffraction ring pattern corresponding to the random orientation. Combining the phase, structural and morphological information suggested that the DL-alanine MFs were grown along the [001] polar direction (Fig. S4). The branching and merging most likely occurred along the side facets that were likely composed of the (110), (210) or (410) facets, where a rich amount of hydrogen bonds on those facets favorites the splitting and joining interactions. The (110), (210) or (410) side facets could be verified by measuring the angles between these facets and top/bottom side of MFs and comparing them with theoretical angles between the crystal facets.

It is known that amino acid nano/micro-structures are typically self-assembled through a supramolecular packing process governed by interfacial hydrogen bonding, which usually led to a straight fibrous morphology[45,46]. We discovered that adjusting the solvent-molecular interaction, combined with surface tension tuning, could control the bifurcation phenomenon and led the formation of the truss-like network. Similar to other amino acid or peptide systems, our DL-alanine MFs also started with supramolecular building blocks, a rod-like nanofibril (Fig. S5). In the vertical precipitation system, the capillary and Marangoni flow drove the DL-alanine building blocks to the meniscus. Electrostatic interaction and hydrogen bonding between the amino and carboxyl groups directed their oriented attachment into well-faceted MFs (Fig. S5). It is also known that additional amount of alcohol molecules, such as ethanol, could preferably bond to the carboxyl groups on amino acid crystal surfaces, and thus limits their further interaction with incoming nanofibril building blocks[47,48]. As illustrated in Fig. 1g, this selective molecular interaction induced a preferred growth direction, i.e. along the [001] for MF precipitation. Meanwhile, the weakened interaction on the side facets also led to a less stable attachment of the nanofibril building blocks along their side

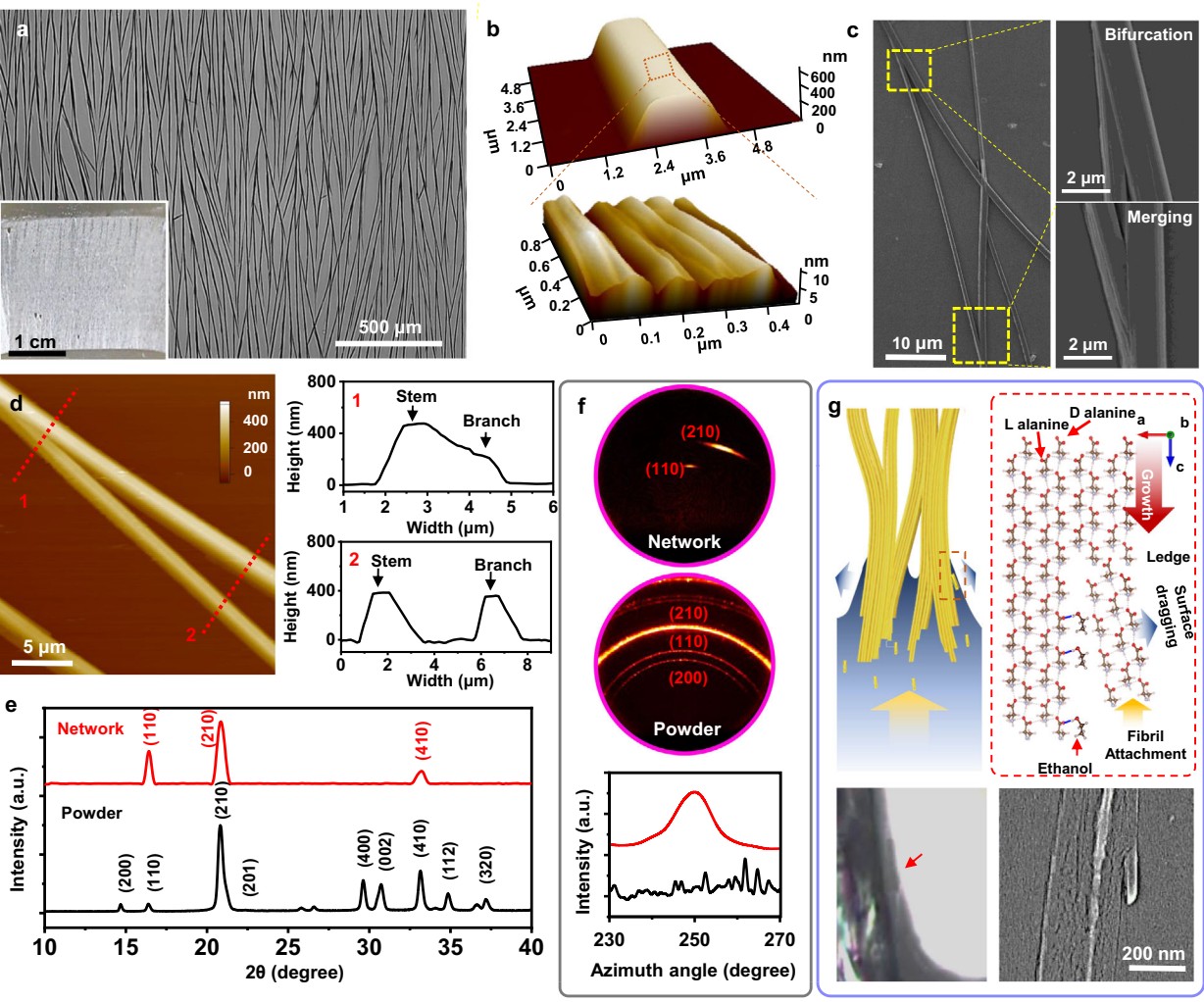

**Fig. 1 | Synthesis and growth mechanism of piezoelectric DL-alanine biocrystal network. a** Large-area optical microscope image of a DL-alanine truss-like network. Inset is a digital photograph of the network grown on a Si wafer substrate. **b** AFM topography images of a single DL-alanine MF (top) and a zoomed-in image showing the fibrous surface feature (bottom). **c** SEM image of an intersection area including both bifurcation and merging features of the MF network. **d** AFM topography image of a bifurcation region. Right-panel compares the height profiles of two MFs at the bifurcated region (top) and after bifurcation (bottom) as marked by the dashed lines in the AFM image. **e** XRD spectra of as-prepared truss-like DL-alanine MF network (red) and DL-alanine raw powders (black). **f** 2D XRD of DL-alanine network (top) and DL-alanine raw powders (bottom). **g** Schematics illustrating the branching process of DL-alanine MFs driven by solvent-molecular interaction and surface tension. L alanine and D alanine molecules are indicated by red arrows. Bottom left inset: optical microscope image showing the small contact angle at the DL-alanine MF and water interface. Bottom right inset: SEM image showing a MF with a developing branch.

walls compared to the growth front. As a result, when the growth front was disturbed by other factors, such as surface tension or interfacial roughness that may introduce a lateral force, bifurcation could be induced (inset of Fig. 1g).

This bifurcation mechanism was further evaluated by varying the ethanol-water mixing ratio, which changed the surface tension at the MF-solution interface as well as the amount of ethanol molecules in solution (Fig. S6). The non-linear change of contact angles under different ethanol/water ratios is mainly due to the huge difference of surface tension between water and ethanol. Once the ethanol to water ratio is over 6, the surface tension of the solution is very close to pure ethanol and can completely wet the DL-alanine MF. Regular bifurcations were obtained at the ethanol-to-water ratio of 3:1-4:1, where a balanced molecular force and surface tension may be reached. A higher water content (2:1) yielded mostly straight MFs with a less regular bifurcation phenomenon, suggesting the side wall interaction may still be too strong to be regularly disturbed by the interfacial tension (Fig. S7). Further increasing the ethanol concentration led to curved MF formation, possibly as a result of the very low surface

tension. Under this situation, the extremely small contact angle may not provide enough space or pulling force to induce a new branch formation. However, it may still slightly deflect the attachment nano-fibrils and shift the MF orientation gradually during precipitation forming a curved geometry macroscopically. Similarly, surface roughness could also change interfacial interaction that resulted in similar bifurcation variations (Fig. S8). An ideal truss-like network was obtained on a very flat surface with a surface roughness ($R_q$) of ~8 nm, where the control of bifurcation was dominated by the solid-liquid interfacial tension. Higher surface roughness increased the interface instability, and thereby led to much denser and more irregular branches. Understanding of the branching mechanism allowed selective distribution of the branching location and thus to control the DL-alanine pattern. For example, creating fine trenches on a flat surface could further define the branching distributions and density (Fig. S9).

## Stretchability of DL-alanine crystalline network

This strategy is fairly versatile and can be achieved on a large variety of hydrophilic substrates, including ceramics (e.g., glass), metals (e.g.,

gold), and polymers (*e.g.*, Poly(methyl methacrylate)) (Fig. S10). Such a truss-like network is a common geometry used in many packaging materials to provide a large stretchability (middle-right inset of Fig. 2a). Thus, a stretchable DL-alanine piezoelectric film was fabricated on a polydimethylsiloxane (PDMS) elastomer thin film treated by oxygen plasma to improve its hydrophilicity. A uniform truss-like DL-alanine MF network was obtained with similar fiber sizes and mesh dimensions as those grown on Si over the entire inch-level surface area (Fig. S11). An additional PDMS layer was applied to encapsulate the as-grown DL-alanine network, and the entire film was released from the supporting substrate as a free-standing system for further characterizations (Figs. 2a and S11). In this truss-like microstructure, large tensile strains along different directions could be compensated by opening or closing the truss meshes, which minimized the strain directly implemented to the crystal lattices (bottom inset of Fig. 2a). Although different substrates may influence the stretchability of network itself by

varying branching angles and density, the overall stretchability of a network-substrate system is only meaningful on stretchable substrates like PDMS.

To demonstrate this omnidirectional stretchability, we first examined the structure of DL-Alanine MF network when it was stretched along both longitudinal (MF growth direction) and transverse directions (perpendicular to the MF growth direction) (Fig. S12). Transverse stretching obviously widened the truss units without showing any signs of fracture or detachment of the MFs (Figs. 2b and S13). A typical angle between two MLs expanded from 18.3° to 24.4° as the transverse tensile strain increased from 0 to 40%. When stretched longitudinally, the film elongation was compensated by closing the open meshes with narrowed intersection angles (Figs. 2c and S14, Supplementary Movie 1 and Supplementary Movie 2). Fractures of MLs were detected when the longitudinal strain exceeded 20% (Fig. S14). The different stretchability along the two directions was a result of the

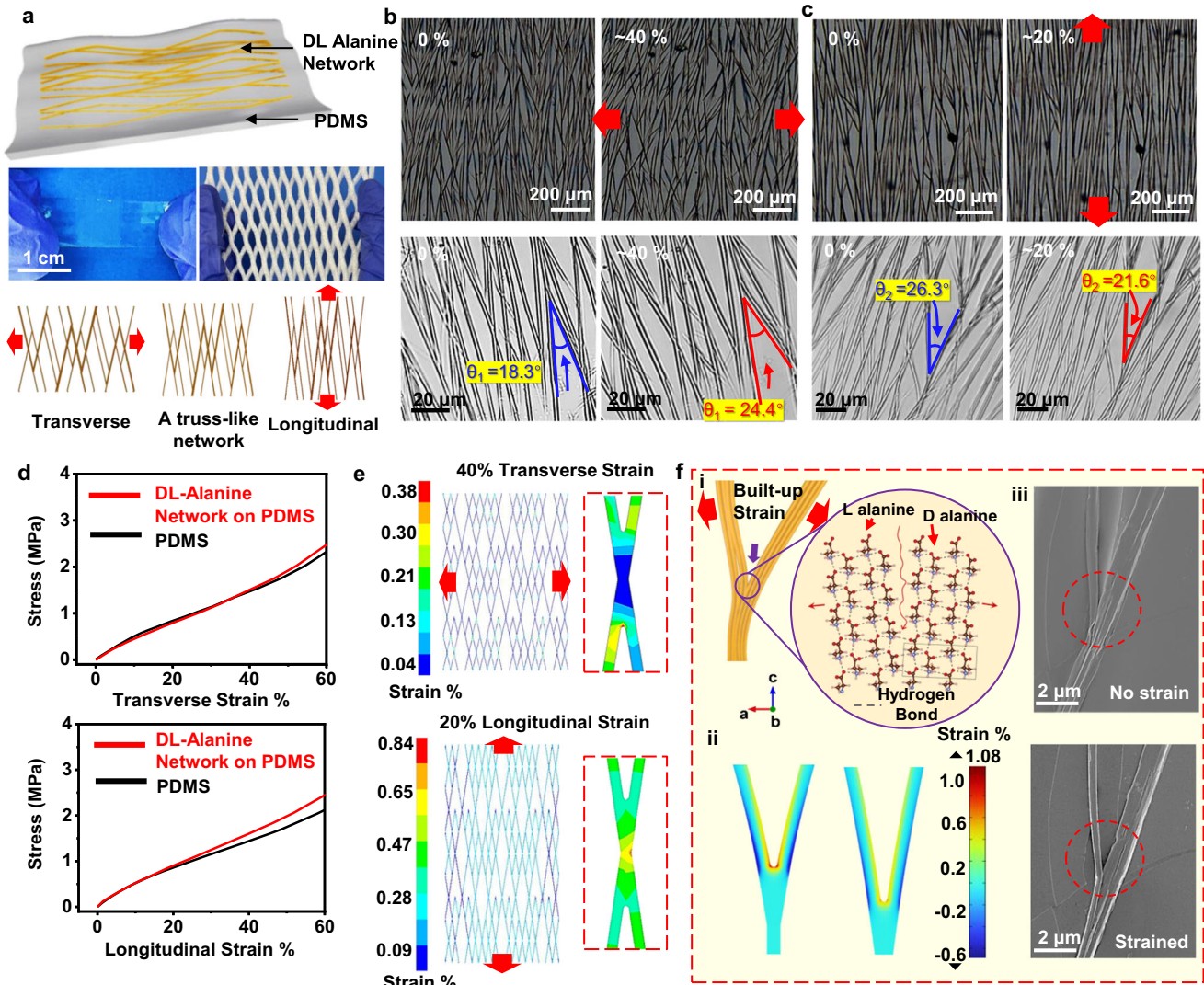

**Fig. 2 | Stretchability of DL-alanine crystalline network. a** Schematics of a free-standing stretchable DL-alanine network film supported by PDMS (poly-dimethylsiloxane) elastomer. Top: DL-alanine network grown on a PDMS film. Middle: a truss-like network of package materials showing the principle of structure-enabled stretchability. Bottom inset: Schematic illustrating the tensile strains resulted from the opening and narrowing of truss units. **b** A DL-alanine network at 0% (left) and 40% (right) transverse tensile strain. Bottom enlarged images reveal a typical joint angle increase. **c** A DL-alanine network at 0% (left) and 20% (right) longitudinal tensile strain. Bottom enlarged images reveal a typical joint

angle decrease. **d** Stress-strain curves of a PDMS-supported DL-alanine network and a PDMS film under transverse (top) and longitudinal (bottom) tensile strains. **e** FEA simulations of strain distribution on simplified MF networks under 40% transverse (top) and 20% longitudinal (bottom) tensile strains. Right insets are zoomed-in views showing strain distributions at intersections. **f** Schematics (i), simulations (ii), and SEM observations (iii) of releasing built-up strain energy by breaking hydrogen bonding between two joining MFs. L alanine and D alanine molecules are indicated by red arrows.

elongated truss-like pattern that allowed more room for transverse expansion as well as the excellent flexibility of the DL-alanine MFs. This high stretchability demonstrated by the truss-like network is more than 20 times higher than the maximally allowed strains in amino acid, peptide or protein crystals (fracture limit ~ 1%)[49], and even higher than those of piezoelectric ceramics[50,51]. Fatigue test of DL-alanine network on PDMS was performed by applying cyclic strains. Only a very few dark-contrast areas appeared after 100 tensile straining cycles, suggesting the DL-alanine/PDMS film had a good stability with consistent stress-strain behavior over 100 large cyclic strains (Fig. S15). It is also important to point out that the tensile moduli along both directions remained nearly the same as that of the bare PDMS film (3.8 MPa) within the elastic region (Fig. 2d). This could be attributed to the fact that only small bending of MF was involved within this stretchable range. Obvious moduli increase was observed when tensile strain exceeded this range as fracture started to take place. Besides, the DL-alanine network had a strong interfacial interaction with the PDMS substrate, which was proved by the removal test using a 3 M very high bond (VHB) tape (Fig. S16). The strong interfacial bonding ensured consistent adhesion without observable buckling of MFs within the stretchable range. Small portions of detachment were only observed at exceeding large transverse tensile strains ( > 45%) (Fig. S13d, e).

The truss-like structure-enabled omnidirectional stretchability was further explained by finite element analysis (FEA) simulation using a simplified model that consists of regular quadrilateral units with two sets of sizes (200 μm long and 25 μm wide; 400 μm long and 50 μm wide) on PDMS to emulate the actual structure. Simulation revealed that a 40% transversal stretching widens the mesh units and only produces 0.38% peak tensile strains at the junction regions (Fig. 2e). Similarly, a 20% longitudinal stretching narrows the mesh units and only yielded 0.84% peak tensile strains at junctions. Simulation of dynamic strain distribution revealed that the strains always concentrate on the same location of "X" junction during the unidirectional tensile straining (Fig. S17a). As piezoelectric responses are directly correlated to strains, the simulation of piezoelectric potential demonstrated that the highest piezoelectric response is near the "X" junction area while the majority of the network has a low piezoelectric potential, which is consistent with the simulated strain distribution (Fig. S17b). In addition, the concentrated local strain at the junction could be further reduced by opening up more junction area between the MFs. Since the joining facets are primarily bonded via intermolecular hydrogen bonding, the concentrated stress may overcome the hydrogen bonding strength and further pull the two jointing MFs apart to release the built-up strain energy (Fig. 2f–i). FEA simulation showed that opening the branch junction by additional 10 μm could lower the local strain by more than 10% relatively (at 20% stretching) (Fig. 2f-ii). This unique strain-releasing feature was observed at a strained intersection, where a branch was opened by additional ~2 μm when subjected to 20% stretching (Fig. 2f-iii).

## Piezoelectricity and biocompatibility of DL-alanine crystalline network

The excellent stretchability and structural integrity allowed the DL-alanine crystal network to show a strong piezoelectricity under various straining conditions. The piezoelectricity of individual DL-alanine microfiber was first characterized by piezo-response force microscopy (PFM). A large PFM signal amplitude was obtained from the microfiber surface (Fig. 3a-i), suggesting a strong out-of-plane polarization. The scan exhibited a uniform phase response over the fiber surfaces, indicating well-aligned dipole distribution throughout the entire fiber. PFM amplitude and phase with a similar level of contrast intensity were also observed from the junction areas (Fig. 3a-ii and iii). Although small amplitude variations were discovered from the two MFs due to their small size difference, their phase contrasts were much more uniform, suggesting all MFs share the same polarization orientation, before and

after their bifurcation or merging. The piezoelectric coefficient $d_{33}^{eff}$ of single DL-alanine MFs can be assessed by amplitude responses under different driving voltages (Fig. S18). Since DL-alanine crystals has non-zero $d_{33}$ and the transverse and shear piezoelectric components in addition to the longitudinal component may contribute to the vertical PFM response due to the shape and geometry of DL alanine MFs, the out of plane direction is thus defined as the effective "3" direction together with quantified piezoelectric coefficient as effective $d_{33}$ rather than assigned to a specific tensor element in the coefficient matrix. Calibrated by standard LiNbO$_3$ sample (Fig. S19), this DL alanine MF revealed a $d_{33}^{eff}$ of ~5.5 pm. The measured $d_{33}$ was reasonably smaller than the calculated value of 10.34 pm/V from a perfect structure[39], given the practical measurement conditions in PFM with tip-sample interactions and structural imperfection.

This well aligned polarization is essential for the DL-alanine MF network to show macroscopic piezoelectricity, which could be evidenced by second harmonic generation (SHG) microscopy. Fig. 3b shows an unpolarized SHG image of a large-area DL-alanine network supported on a PDMS film, where a very uniform signal intensity could be visualized across the entire structure. The polarization angle of SHG was then adjusted from 0° to 90°. The intensity of SHG signal decreased monotonically as the polarization angle increasing (Fig. 3c) following the Neumann's principle (Supplementary Text)[52,53]. The SHG signal variations kept a good uniformity and consistency over the entire network under all polarization angles, validating the well-aligned dipoles at the macroscopic scale.

The excellent structural integrity under large strains allowed the truss-like MF network to preserve its strong macroscopic piezoelectricity when stretched along different directions, an essential capability for an omnidirectionally stretchable material. The in-plane piezoelectricity under various tensile strains was characterized by measuring the current output in response to a normal tapping force ( ~3 N) at a frequency of 1 Hz, when the network was stretched to 20% laterally along a series of directions from 0° to 90° (0° was defined as the MF growth direction) (Fig. 3d). A 3 N tapping force was applied by pressing the PDMS substrate with another PDMS rod. The PDMS rod remained a close contact with the PDMS substrate during the cyclic pressing to minimize the triboelectric contribution (Fig. S20). The near zero outputs of pure PDMS substrate without DL-alanine network further confirmed the negligible triboelectric contributions to the piezoelectricity measurements. Constant 3 N tapping force under different strains led to peak-to-peak voltage output of ~90 mV without significant variations, whereas the voltage output increased significantly from ~30 mV (1 N) to ~150 mV (6 N) when larger tapping forces were applied (Fig. S21). Under longitudinal stretching (0°), the MF film generated 0.609 ± 0.117 nA, equivalent to an effective in-plane piezoelectric coefficient |$d_{13}^{eff}$| of 5.72 ± 1.64 pC/N. High accuracy of piezoelectric coefficient estimation based on low current measurement by preamplifier can be achieved by optimizing the test conditions (Methods), which have been demonstrated by previous works[54,55]. The effective piezoelectric coefficient $d_{13}^{eff}$ of the entire DL-alanine network film is defined as the in-plane polarization resulted from vertical displacement (Supplementary Methods). While the measured effective piezoelectric coefficient is the transverse $d_{13}$ of the whole DL alanine network based on the definition, longitudinal component (effective $d_{11}$) of the whole film may also have small contributions to the measured piezoelectric response due to the in-plane stretching of the network under tapping force. Similar current outputs (and corresponding |$d_{13}^{eff}$|) of 0.477 ± 0.119 nA (4.20 ± 0.96 pC/N), 0.550 ± 0.120 nA (6.21 ± 1.65 pC/N) and 0.587 ± 0.118 nA (5.12 ± 1.22 pC/N) were obtained as the stretching angle increased to 30°, 60° and 90° (transverse stretching), respectively (Fig. S22). Without strain, the device also had a similar level current output of ~0.490 ± 0.102 nA, corresponding to 4.32 ± 1.06 pC/N. The estimated values of 4-6 pC/N are consistent with piezocoefficient results of DL-alanine crystals from

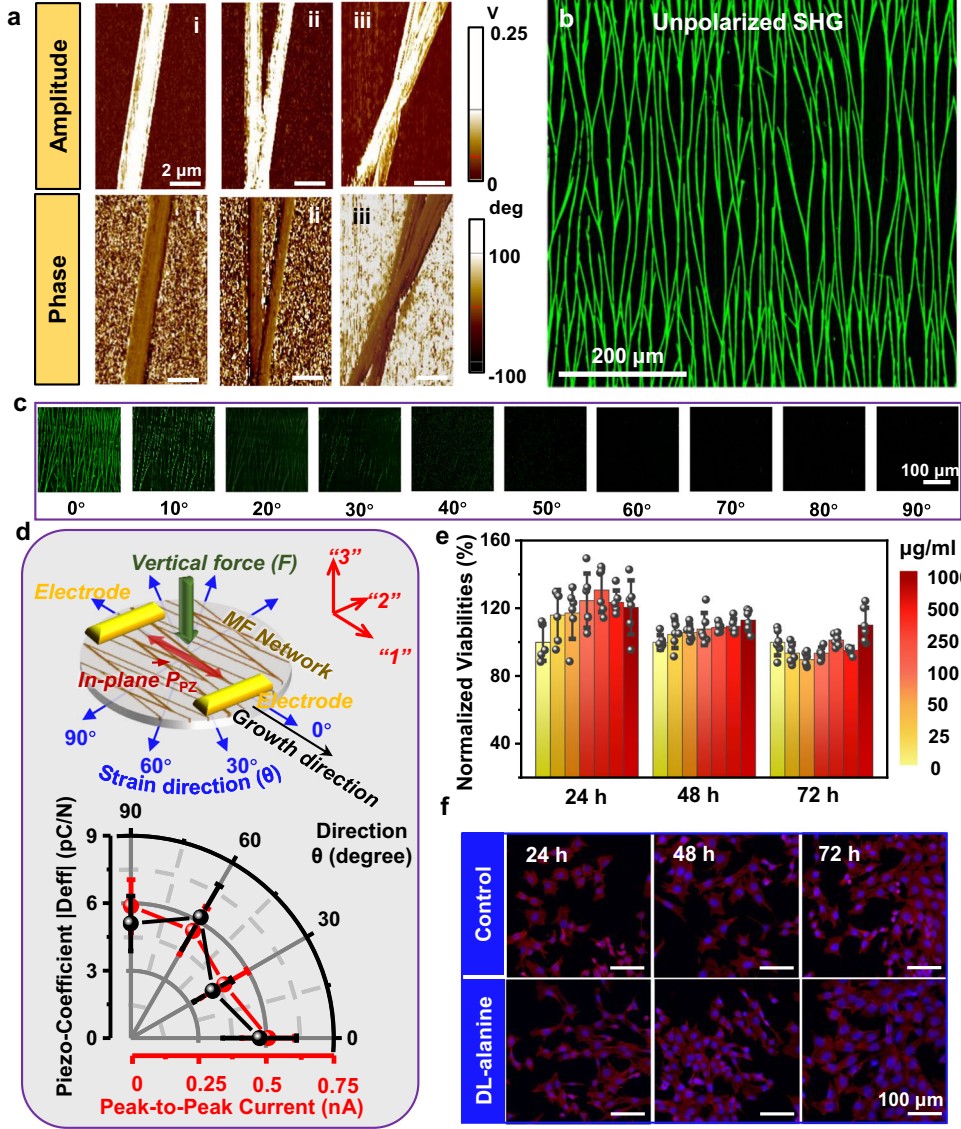

**Fig. 3 | Piezoelectricity and biocompatibility of DL-alanine biocrystal network.** **a** PFM amplitude (top) and phase (bottom) responses of a single MF (left), bifurcated MFs (middle), and a MF cross (right). **b** Large-area unpolarized second harmonic generation (SHG) image of DL-alanine network showing uniform polarization contrast. **c** Polarized SHG images of the MF networks under excitation linear polarization orientation varying from 0 degree to 90 degrees. **d** Current output and in-plane effective piezoelectric coefficient of the MF network under various tensile strains in response to a 1 Hz and 3 N normal tapping force applying at the center. Electrode configuration and effective directions are indicated in the

top schematic figure. $n = 5$ for each group. The error bars represent standard deviations. All data in Fig. 3d are presented as mean ± s.d. **e** Quantitative cell viability analysis and comparison during a 3-day culturing period. $n = 6$ for each group. The error bars represent standard deviations. All data in Fig. 3e are presented as mean ± s.d. Colors represent the concentration of DL alanine in culture media solutions. **f** Fluorescence microscopy images showing the normal morphology evolution of cells cultured in a DM solution with various amounts of DL-alanine dissolved inside during a three-day period.

previous studies[39,56]. The stable piezoelectric performance under large strains along different directions evidenced the excellent omnidirectional stretchability in terms of both structure and property preservation. Several factors can affect the piezoelectric responses of the network. Larger branching angles and misorientations might result in less net in-plane polarization along the growth direction, resulted in variations of in-plane piezoelectric responses of the DL-alanine network. However, the branching and orientation should not affect the net out-of-plane polarization and the out-of-plane piezoelectric responses of the network. The density of DL alanine MF may also affect the resulting current. Nevertheless, the density-dependent piezoelectric output would be complicated. On the one hand, higher density of DL-alanine MFs increased the number of MFs that contribute charge and improve the current output. On the other hand, higher density

may also reduce the strains distributed in each single MF, and therefore lower piezoelectric charges and the current output of whole network.

As a natural biomaterial, the DL-alanine microstructure is also biocompatible and biodegradable. To demonstrate this property, we cultured mouse vascular smooth muscle cell (MOVAS) in Dulbecco's modified Eagle's medium (DMEM) solutions with various amounts of DL-alanine dissolved inside (0–1000 µg/ml). Quantitative analysis revealed that the cell viabilities (normalized to control group (0 µg/ml)) at different concentrations all remained above 100% during the 3-day period (Fig. 3e). Immunofluorescence staining performed over the same period further showed that the MOVAS cells exhibited normal behavior and reached a higher density with a typical filamentous and stretched morphology on days 2 and 3 (Fig. 3f). The cell

morphologies, distributions, and densities did not show any significant differences between the control group and experimental groups. Both results evidenced that the as-synthesized DL-alanine ML network has no cytotoxicity. Once exposed to a phosphate buffered saline (PBS) solution, the entire MF network would completely dissolve in one minute (Fig. S23), indicating its potential to serve as a biodegradable functional component.

## DL-alanine network based wearable and implantable electromechanical devices

The unique combination of softness, omnidirectional stretchability and biocompatibility allows the DL-alanine MF network to be used for wearable and implantable electromechanical devices with tissue-compatible mechanical behavior. To demonstrate this capability, an omnidirectionally stretchable piezoelectric nanogenerator (NG) was fabricated by integrating the DL-alanine network with a stretchable electrode made from percolated silver (Ag) nanowires (NWs) (Figs. 4a and S24). The Ag NW-based electrode was also supported on a PDMS substrate (upper right inset of Fig. 4a). It could retain a very low resistivity of <50Ω under strains up to 25% in all directions (Fig. S24). The Ag NWs were placed in direct contact with the DL-alanine MFs to

ensure effective collection of piezoelectric responses. The single electrode configuration is desired to realize a fully stretchable device, in order to minimize the strain-related conductivity change in other two-electrode configurations. The complete NG device was soft, thin, and able to bear large twist deformations (lower right inset of Fig. 4a). The energy generation performance was characterized by measuring the voltage output between the Ag electrode and ground in response to a low-frequency (1 Hz) pressure oscillation (~156 kPa). Meanwhile, the NG was stretched to 20% along the same series of directions from 0° to 90° (Fig. 4b), The peak-to-peak voltage output of ~90 mV was obtained on an external load of 100 MΩ from all strain directions (Fig. 4c). These values also matched the voltage output (80 mV) measured without introducing any lateral strain. The relatively stable piezoelectric output along different stretching directions suggested that the DL-alanine-based NG device could function appropriately under random stretching directions that can often be found in biological systems.

The omnidirectional stretchability allows the NG to conform to skin or tissue surfaces and to be responsive to irregular mechanical motions. For example, it could serve well as a piezoelectric tactile sensor particularly for body locations that often subject to large

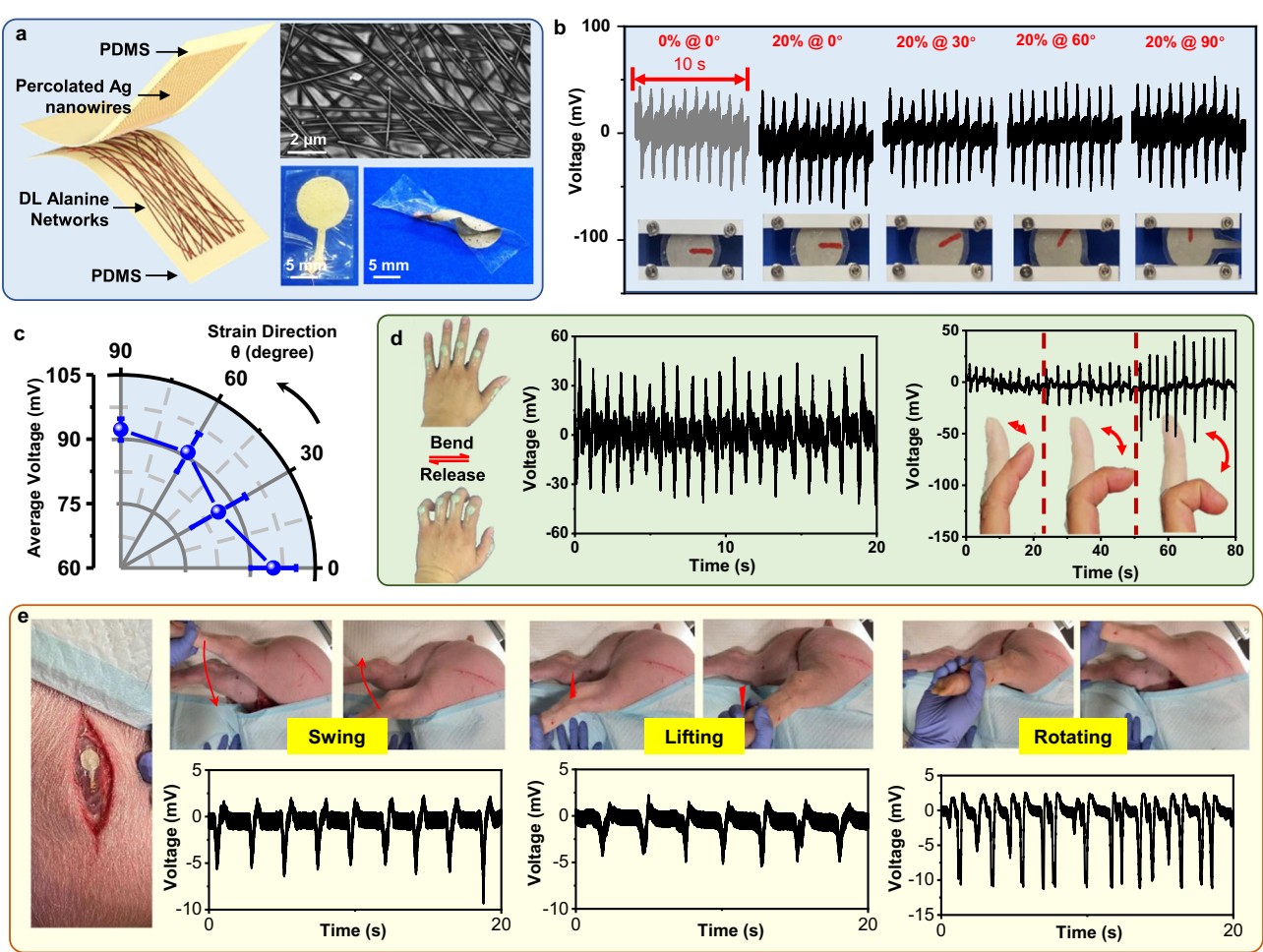

**Fig. 4 | Wearable and implantable electromechanical devices with tissue-mimicking stretchability. a** Schematics of an omnidirectionally stretchable piezoelectric NG fabricated by integrating the DL-alanine network with a stretchable Ag NWs electrode. Top inset: a SEM image of percolated Ag NWs. Bottom inset: digital images of a stretchable piezoelectric NG. **b** The voltage outputs measured from the stretchable NG without strain and with 20% tensile strains along various directions in response to 1 Hz pressure oscillation (~156 kPa). **c** A plot of average peak-to-peak voltage outputs as a function of the strain direction. n = 5 for each

group. The error bars represent standard deviations. All data in Fig. 4c are presented as mean ± s.d. **d** The conformal attachment of a stretchable NG on human hand knuckles when fingers were bent and released (left). The stable voltage output under repeating finger bending (middle), and different voltage outputs in respond to different degrees of finger bending (right). **e** Implantation of a stretchable NG on the top thigh muscle of a swine (left), and its in vivo piezoelectric voltage outputs under different leg movements: (i) swing; (ii) lifting; (iii) rotating.

deformations, such as knuckles and wrist. After being strained 20%, this piezoelectric tactile sensor based on DL-alanine network was still able to generate consistent voltage signals with no significant reduction in amplitude over 1500 cyclic bending, revealing excellent durability (Fig. S25). This device could be seamlessly attached to the knuckles without delamination under repeating large-angle bending, and produced a stable peak-to-peak voltage output of ~ 60 mV (Fig. 4d). The voltage outputs were well correlated with the degree to bending, demonstrating the unique piezoelectric sensing capability in response to the amplitudes of displacement (Fig. 4d and Supplementary Movie 3). Control devices made of commercial PVDF and bare PDMS thin films (Fig. S26) validated the normal function of single electrode configuration and excluded the noise contributions from wire movement or other measurement-related artifacts. In our measurements, potential interference from external signals due to electromagnetic field or triboelectricity were minimized by pre-amplifier filtering or minimizing contact-separation electrification, respectively, in the measurement of piezoelectric tactile sensor signals. The stretchable NG was also affixed to the wrist conformally. When bended or twisted along random directions, it was able to provide consistent piezoelectric outputs in responses to gentle touching, suggesting the potential as skin sensory receptors for body proprioception (Fig. S27). The energy conversion efficiency η of DL-alanine network/PDMS film can be estimated by comparing the input energy and output energy. An estimated efficiency of approximately 1.08% is comparable to the energy efficiency of previously reported conformal piezoelectric PZT nanoribbon device (1.76%)[41]. To further demonstrate its in vivo performance, the NG device was encapsulated by an additional layer of PDMS and implanted on the outer surface of a swine's thigh muscle (Fig. S28), where very large strains along multiple directions are typically experienced as the leg moves. Different stains were applied by lifting the leg up and down, swinging the leg back and forth, and rotating the leg. The device responded to all movements with steady peak-to-peak voltage outputs of ~10–20 mV, which were well correlated to the amplitude of muscle movements (Fig. 4e). These results validated the intriguing potential for applying the DL-alanine network as an implantable piezoelectric device with tissue-mimicking mechanical properties. Although other piezoelectric materials such as PVDF, PLLA, and PZT composites have been demonstrated as wearable and implantable sensors, this work distinguishes from them by enabling a self-assembled piezoelectric DL-alanine crystal network with omnidirectional stretchability, which provides a critical component for achieving fully stretchable and conformal implantable piezoelectric devices. Future research should enable similar biodegradability and omnidirectional stretchability in electrodes and substrates in order to realize the envisioned devices. Particular efforts to engineer biodegradable elastomers such as poly-(glycerol sebacate) (PGS)[57] and poly-(glycerol dodecanedioate) (PGD)[58] as growth substrates for DL alanine network can greatly facilitate the realization of a fully biodegradable stretchable tactile sensor.

## Discussion

In summary, this work reported a large scale piezoelectric bioorganic crystal network that shows omnidirectional stretchability with tissue-mimicking mechanical properties and a stable piezoelectricity. The ultra-high stretchability is enabled by a truss-like microstructure formed by regularly interconnected DL-alanine MFs. The selective molecular-solvent interaction assisted with tunable surface tension at the growth front was discovered as the major mechanism that drove the self-assembly of MF network with a self-aligned orientation. The truss-like MF assembly was able to withstand large in-plane tensile strains by truss pattern deformation, while well preserved the overall lattice continuity and structure integrity. Therefore, the as-received

inch-scale films were able to endure up to 40% tensile strain while still showing unimpaired piezoelectricity. Combining with the excellent biocompatibility from its biomaterial nature, the piezoelectric DL-alanine MF films enabled the creation of wearable and implantable piezoelectric devices offering tissue-mimicking mechanical property and electromechanical coupling functions. By integrating with stretchable electrodes, an implantable NG was demonstrated that allowed conformal attachment to skin and tissue surfaces, and generating stable electrical outputs in response to multi-directional large motions in vivo. This work provides a scalable solution that introduces omnidirectional stretchability to a piezoelectric biocrystal film, laying a cornerstone for creating tissue-compatible implantable piezoelectric devices for sensing, monitoring, therapeutics and energy harvesting.

## Methods

### Growth of DL-alanine microfiber (MF) network on hydrophilic substrates

DL-alanine powders (Santa Cruz Biotechnology, Inc.) were dissolved in a biphasic solution composed of 20% deionized (DI) water and 80% ethanol (200 Proof). The solution was sonicated in an ultrasonic bath for 1 h at room temperature (25°C) to enable homogeneity. 15 ml DL-alanine solution (0.5 mg/ml) was then added into a 24 ml glass vial (28 × 70 mm, VWR International) to prepare for growth. By placing the glass vials in a customized growth chamber (Fig. S1) with a controlled humidity (10%–15%), air flow, and temperature (23°C), hydrophilic substrates immersed in the solution was vertically pulled out at a very slow speed of 25 μm/min by a modified syringe pump (Harvard Apparatus, Harvard Bioscience, Inc.). The evaporation rate of solution was set to be as low as ~55 mg/h by controlling the air flows inside the growth chamber.

### Growth of DL-alanine microfiber (MF) network on elastomers

Poly(methyl methacrylate) (PMMA) (Mw ~97000, Sigma Aldrich) was dissolved in a chloroform benzene (CB) solution (Anhydrous 99.8 %, Sigma Aldrich). The PMMA solution (1 wt.%) was first spin coated on a glass slide at a speed of 3000 rpm for 30 s. Afterwards, the glass slide was backed in a thermal oven at 80 °C for 10 min to evaporate all remaining CB solution. The polydimethylsiloxane (PDMS) (Sylgard 184, Tow Corning) solution consisting of pre-mixed elastomer and crosslinker at the ratio of 10 to 1 was then spin coated on the glass slide with a thin layer of PMMA on top. The spin coating speed of PDMS was set at 1000 rpm for 120 s. To accelerate the curing process, the spin-coated PDMS was then backed in a thermal oven at 75°C for 3 h. Further, surface of fully cured PDMS was treated by oxygen plasma (100 W for 200 s) in a plasms etching system (PlasmaEtch PE-200). After surface modification, the PDMS/PMMA/Glass substrate was immersed into the DL alanine solution (0.5 mg/ml), and then was vertically pulled out at a constant speed of 25 μm/min. The evaporation rate of solution was set to be as low as ~55 mg/h by controlling the air flows inside the chamber. To make a freestanding MF network/elastomer film, the PDMS/PMMA/Glass substrate was immersed in an acetonitrile (Anhydrous 99.8%, Sigma Aldrich) solution to remove the PMMA in between. Once the PMMA removed, the MF network/PDMS would spontaneously detach from the glass slide.

### Characterizations

**Morphology characterizations.** Microscopy images were obtained with a CCD camera mounted on a Phase Imaging Microscope (DM4 M, Leica), and recorded through imaging capture software. Scanning electron microscopy (SEM) observations of the MF networks and their cross-sections were performed on a Zeiss LEO 1530 field-emission microscope. AFM topography images were obtained using a non-contact probe (PPP-NCHR, Park Systems, South Korea) on a Park Systems XE-70 multimode atomic force microscope.

**Structure analysis.** X-ray diffraction patterns of DL-alanine powders and MF networks were acquired from the Bruker D8 Discovery with Cu Kα radiation.

**Mechanical property characterizations.** The stress-strain curves of glycine-PVA films were measured by a Rheometrics Solids Analyzer III dynamic mechanical analyzer. All the films had the same dimension of 2 cm × 1 cm × 0.01 mm. The stretching of DL-alanine MF network under optical microscope is through a customized tensile platform (Fig. S12).

**Piezoelectric property characterization.** Piezoelectric force microscopy (PFM) measurements on individual DL-alanine MF were performed using the dynamic contact electrostatic force microscopy (DC-EFM) mode on a Park Systems XE-70 multimode atomic force microscope (AFM) and Stanford Research 830 (SR830) lock-in amplifier (LiA). To measure the effective piezoelectric charge coefficient ($|d_{eff}|$), a gentile force of 5 N at 1 Hz was applied at the center of the piezoelectric network/elastomer films with two ends of MF network contacting two in-plane electrodes. The force was quantified by a portable sensor measurement system (compression piezoelectric sensor (CL-YD303) integrated with a four-channel dynamic signal acquisition module (NI 9234) and compact data acquisition chassis (NI, cDAQ-9171)). The corresponding piezoelectric current outputs of the DL alanine network were recorded by connecting probes of a low-noise current preamplifier (Stanford Research Systems, model SR570) connected with LabVIEW system in computer to the in-plane parallel electrodes. To maximize the accuracy of measurement, low noise mode, low input offset current and low-pass filter in the preamplifier are adapted to remove noise from the measurement signal. Furthermore, a customized metal box as shielding is harnessed which can reduce distortion caused by external sources. Second harmonic generation (SHG) imaging was carried out using an upright multiphoton microscope (Nikon A1R high definition (HD) multiphoton confocal microscope). The SHG signals of film samples were measured under 1040 nm radiation from built-in lasers. The polarized SHG signals were measured by adding a polarizer between the laser and the sample.

### Device fabrications and characterizations

The stretchable electrode was made by transferring percolated silver (Ag) nanowires (NWs) on top of PDMS substrates[59]. The Ag NW solution (XFNANO, 20 mg/mL) was spin-casted on Si substrate and dried at 70 °C for 5 min. The PDMS solution consisting of crosslinker and elastomer with 1:10 volume ratio was spin-coated for 40 s onto Ag NWs electrodes. The PDMS was cured for 4 h at 70 °C, and then the percolated Ag NWs were partially embedded into the PDMS matrix to form a stretchable electrode. The Ag NWs/PDMS stretchable electrode was then peeled off from the Si substrate. The conductivity of stretchable Ag electrode under strains was measured by a multimeter (DMM 6500, Keithley). The piezoelectric network/PDMS film was made in close contact with the percolated Ag NWs/PDMS film, and they were further packaged by additional top and bottom PDMS encapsulation layers (without fully cured) through lamination. The whole device was then placed in a thermal oven (45 °C) for curing for 10 h. The voltage outputs of the piezoelectric device with/without strains were measured by connecting two probes of a low-noise voltage preamplifier (Stanford Research Systems, model SR560) into the single electrode and ground, respectively.

### Biological studies

**Cell cytotoxicity.** Mouse vascular smooth muscle cells (MOVAS) were purchased from American Type Culture Collection (ATCC, CRL-2797) and grown as recommended in modified DMEM containing 4.5 g/L D-Glucose (Thermo Scientific, 11965118) supplemented with 10% fetal bovine serum (FBS), 100 U/mL penicillin, and 100 U/mL streptomycin. The cell biocompatibility of DL-alanine MFs was assessed with a CellTiter-Glo assay (Promega, G9242) as described[60]. Different concentrations (25, 50, 100, 250, 500 and 1000 µg/mL) of DL-alanine MFs in culture media solutions were added to a 96-well plate with black wall and clear bottom (Corning 3603). There were three wells in each group. $5\times10^3$ cells were seeded into each well containing 200 ul cell culture medium. After incubation for 48 h at 37 °C and 5% CO$_2$, cell culture medium was aspirated, 100 µL of PBS and 100 µL of CellTiter-Glo solution was added to each well followed by incubation at room temperature for 30 min. Luminescence was recorded on a FlexStation 3 microplate reader (Molecular Devices) at an integration time of 0.5 s per well. The relative cell viability was expressed as (luminescence of sample wells- blank) (luminescence of control wells-blank)×100%, where blank is the luminescence of the wells without cells (PBS and CellTiter-Glo solution only). The cell viability results of each group with different concentration were repeated three times.

**Cell morphology and immunofluorescence staining.** There were four wells in each group. $2.5\times10^4$ MOVAS cells were seeded into each well containing 500 µl cell culture medium with different concentrations of DL-alanine MFs (0 and 1000 µg/mL). After 24, 48 and 72 h, the cytoskeleton and nucleus of the cells were stained with Alexa Fluor 555 Phalloidin (Thermo Fisher Scientific, A34055) and DAPI, respectively. The staining protocol includes following procedures: First, the cell culture medium was discarded, and cells were washed two times with PBS. They were later fixed with 4% paraformaldehyde for 15 min and rinsed three times with PBS. The cell samples were then permeabilized with 0.1% Triton X-100 for 15 min at room temperature and rinsed three times with PBS. The samples were incubated for 30 min at room temperature with 200 µl/well of Alexa Fluor 555 Phalloidin working solution and then washed three times with PBS. Finally, fluorescent mounting medium with DAPI (GBI Labs, E19-100) was added and incubated for 5 min at room temperature before applying cover slip. After staining, the cells were imaged using a Nikon A1RS high definition (HD) confocal microscope.

**Animal experiments.** All animal experiments were conducted under a protocol approved by the University of Wisconsin Institutional Animal Care and Use Committee. Sex was not considered in study design. Briefly, domestic male adult swine (Duroc, Landrace, Large white, ~30 kg, n = 3) were sedated with intramuscular Telazol (4 mg/kg) and xylazine (2 mg/kg) and then intubated and ventilated with a respirator and anesthetized with isoflurane (2%) and oxygen. Ventilation was adjusted to maintain blood gases in the physiological range. Animals were monitored continuously for anesthetic state by jaw tension, heart rate, blood pressure, end-tidal CO$_2$ and oxygen saturation. A skin incision of ~ 5–10 cm was made in the thigh region, and the packaged piezoelectric stretchable device was implanted subcutaneously. The incision was closed and the leg of animal was stretched. The device outputs under strains with different directions was studied. The voltage outputs of implanted piezoelectric device were measured by connecting two probes of a low-noise voltage preamplifier (Stanford Research Systems, model SR560) into the single electrode and ground, respectively.

### Finite element analysis

The strain distribution of simplified MF DL-alanine networks was simulated by the commercial FEA software ANSYS 19.0 Version. The simplified model that consists of regular quadrilateral units with two sets of sizes (200 µm long and 25 µm wide; 400 µm long and 50 µm wide) on PDMS (50 µm). The individual belt of the simplified network has a width of 2 µm and a height of 400 nm. The values of elastic moduli of both DL-alanine and PDMS were set based on the literature[39,61].

## Statistics and reproducibility

For experiments related to morphology characterization in Figs. 1a, c, d, and 4a, representative results are presented from a minimum of n = 10 independently repeated experiments. For experiments related to the measurement of materials property in Figs. 2f, 3a–c and f, representative results are presented from a minimum of $n$ = 3 independent experiments performed. Further information on research design is available in the Nature Research Reporting Summary linked to this article.

## Ethics statement

The research conducted in this study complies with all ethical regulations. All biological-based in vitro and in vivo experiment protocols were approved by the University of Wisconsin. All animal experiments were conducted under a protocol (Protocol ID: M006023) approved by the University of Wisconsin Institutional Animal Care and Use Committee. Approval for conducting human subjects research was obtained from the University of Wisconsin Institutional Review Board (IRB) prior to the start of this project, IRB protocol (ID: 2022-0805). All data measurements on the bodies of participants are performed with their full and informed consent.

## Reporting summary

Further information on research design is available in the Nature Portfolio Reporting Summary linked to this article.

## Data availability

The authors declare that all data supporting the findings of this study are available within the Article and its Supplementary Information. The raw data generated in this study are available from the corresponding author upon request.

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

## Acknowledgements

This work is supported by the National Heart, Lung, and Blood Institute of the National Institutes of Health under Award Number R01HL157077. The content is solely the responsibility of the authors and does not necessarily represent the official views of the National Institutes of Health.

## Author contributions

J.L. and X.W. conceived the idea and designed the research. J.L. and Y.W. performed film synthesis and device fabrication. J.L., C.C., and J.S. carried out mechanical and piezoelectric characterizations. H.Z. and F.Y. conducted mechanical and piezoelectric simulations. J.L., Y.D., and Z.Z. performed morphology and structure characterizations. Z.S. performed the cell toxicity study. T.H. performed the in vivo experiments. Y.M. and X.W. supervised the work. J.L., Y.M. and X.W. analyzed the data and wrote the manuscript. All authors reviewed and commented on the manuscript.

## Competing interests

The authors declare the following competing interest: J.L., and X.W. are inventors on a patent application [P230250US01 (1512.906)] filed through the Wisconsin Alumni Research Foundation. The status of application is pending. The remaining authors declare no competing interests.
