## [Peer Review File · Nature Communications]

REVIEWER COMMENTS

Reviewer #1 (Remarks to the Author):

The authors presented a novel strategy to fabricate piezoelectric thin films made of DL-alanine and its potential application for real-time health monitoring. The stretchability is enabled by the truss-like microstructure formed by interconnected DL-alanine microfibers. In this work, the authors have reported the mechanism in forming the truss-like network of DL-alanine, the network structure, and microfibers topography. The manuscript has some areas to be improved. These include piezoelectric and mechanical characterizations as well as the demonstration of the material application. Overall, the manuscript is well-written and organized. The work is interesting and novel, yet there are several comments for the authors that might be useful to improve the quality of their studies:

1. The authors demonstrated that their strategy could be achieved on a variety of hydrophilic substrates. So, I wonder if the stretchability of the DL-alanine network depends on the growing substrate. It seems that the branding angles between DL-alanine microfibers (which determine the stretchability of the network) are different on different substrates (Figure S10). Please clarify it.
2. For the truss-like structure, it is recommended to the authors perform fatigue characterization, where they can apply cyclic strains to the network. This could be helpful in determining the functional lifetime of the network/device.
3. The authors should investigate the interfacial interaction between the DL-alanine microfibers and the substrate. Will the network be buckling and detached from the substrate under bending?
4. Although DL-alanine has been reported to exhibit piezoelectricity, it is still recommended to have non-piezoelectric control samples (e.g. PDMS) when the author characterizes the piezoelectric coefficient. As the piezo-coefficient reported in the manuscript is relatively small (around 6 pC/N), I am concerned it may be interfering with triboelectricity.
5. In addition, the authors should provide more data regarding its piezoelectric performance. For instance, output voltages under different applied strains/forces or the polarity experiment (swapping electrodes).
6. Please justify what is the piezoelectric coefficient (e.g., d_{31} , d_{33} , or d_{15}) that the authors measured.
7. Can the authors perform PFM to assess the piezo coefficients of single microfiber? Do the branding angles and orientations of the microfibers in the networks affect the overall effective piezoelectric performance of the film?
8. The advantage of DL-alanine is its biocompatibility and mostly biodegradability, which makes it favorable for implantable applications. Yet, the authors utilized their material for a wearable tactile sensor, which diminishes the benefits of the materials, especially since there have been extensive reports of other piezoelectric materials (e.g., PVDF, PLLA, PZT composites, etc.) for wearable sensors with higher performance. The work could be more significant if the authors could grow the DL-alanine

networks on a flexible biodegradable substrate (instead of growing on PDMS) to make the device completely biodegradable and use it to detect the leg movements in swine.

Reviewer #2 (Remarks to the Author):

This manuscript describes self-assembly of DL-alanine possessing truss-like microfiber structures, its piezoelectric and stretchability characterization, and their application on the wearable/implantable setting. Stretchability is a key property for wearable devices to adapt our bodily movement. Piezoelectricity has broad application potentials as a wearable device. However, most of inorganic piezoelectric materials are crystal based so that brittle or not stretchable, and not-biocompatible. In this manuscript, the authors synthesized organic microfiber structure using DL-alanine. The resulting microfiber structures with truss-like network structures can adapt mechanical stretching in an omnidirectional manner. They also fabricate stretchable device using PDMS/Ag nanowire substrates and demonstrated their biocompatible and motion sensing properties. The result is interesting and shows the full spectrum of piezoelectric materials from synthesis, characterization, and in vivo demonstration. However, although the stretchability part might be novel, many components of the observation or characterization are not that novel. In addition, it is limited to attract broad interest. Therefore, this reviewer thinks it is not suitable for Nature Comm. Instead, it might be better suit for more specific journal specialized for nanoscience, energy, or relevant field.

Here are more specific points that need to be addressed further.

- Many amino acids exhibit piezoelectric properties. In this work, the authors used a racemic mixture of D- and L-alanine. The authors need to explain why DL-alanine and how this racemic mixture affect the polarization of the materials. Most of the schematic illustration in Fig 1g and 2f does not indicate or reflect the racemic nature of the materials. Furthermore, does D-alanine, or L-alanine have the same self-assembling properties or exhibit mirror symmetric results in piezoelectric? How racemic mixture of the compound can generate polarized structures and exhibit SFG in Fig
- In Fig 1f, need more description to explain what is 2D XRD and how this 2D XRD experiment was performed. I think the 1f left, right is mis-labelled between up and down.
- Fig s4 schematic: (410), (210) (110) was specified on the one facet. Need more explanation. It might need more data to link the AFM image with 410, 210, 110.
- Fig s6: Contact angle measurement is confusing. By mixing EtOH: H₂O in different ratio, it does not exhibit the consistent contact angle. Need more explanation. Typo? (tensor => tension?)
- Each micrograph needs to indicate where film growth (puling) direction is.
- Fig S9E: Inset height profile need more explanation. Redline seems not match with the profile.

- Is it possible to characterize d_{33} or d_{14} values from the observed microfibers because it is large enough?

- Fig S17: There is a schematic to indicate the charge on +/- . Based on the described structures in AFM images on Fig S4, schematic needs more description how the polarized + and – surface exist.

Reviewer #3 (Remarks to the Author):

The authors present a study regarding the fabrication and piezoelectric properties of peptide-based networks exhibiting high degree of stretchability, while maintaining piezoelectricity, thus allowing conformable and stretchable piezoelectric devices.

I think the paper is very well written, the procedures and results are clearly described and presented. I have a few comments for the authors to address which I believe would improve the related discussion and the readability of the paper:

1. It appears that the main novelty is in the controlled fabrication of the peptide truss and the ability to use it as an electromechanical device. If that is the case, a clearer comparison to some previous work on conformal PE devices is missing (some is already cited), in terms of efficiency, ease of production etc'.
2. Some Figures involve several experiments; I believe it can be confusing and the paper can benefit from their separation.
3. Regarding the three-point bending type current generation (Fig 3). The experimental setup is not well described in the main text. I suggest improving the schematic. Was the measurement always done in the direction of the fibers? It is not clear. In that regard - what were the conditions of current recording? For low current sensitivities (as were probably used), a high input resistor affects the current and therefore the direct inference of effective d coefficient is not straightforward (it is no longer short-circuit current). I urge the authors to examine the conditions used and analyze the experimental data accordingly.
4. Which d coefficient is dominant in the measurements? Is it d_{33} ? How do the calculated values (or corrected ones) compare to the expected d_{33} ? In fact, I'm not sure the paper suggests a value for the piezoelectric coefficients.
5. Why isn't there a value calculated from AFM? Why wasn't lateral AFM conducted? I suggest this can add to the value of the results.
6. How does the density of fibers affect the resulting current?
7. I am very curious regarding the piezoelectric performance at highly stretched states. Is the AC mechanical loading concentrated in different locations than the steady state stretch? I believe that on

top of the mechanical FE simulations, which are very helpful in asserting the authors' understanding of the mechanical behavior, piezoelectric simulations can be carried out and greatly improve the complete understanding of the device.

8. Similar to point 1 - how does the flexible energy harvester compare in efficiency to others reported in literature?

9. Why was a single-electrode configuration used? is it less efficient than a "regular" two electrode configuration? Also here, the schematic should be included in the main text.

I look forward to seeing the revised manuscript.

Reviewer #1 (Remarks to the Author):

The authors presented a novel strategy to fabricate piezoelectric thin films made of DL-alanine and its potential application for real-time health monitoring. The stretchability is enabled by the truss-like microstructure formed by interconnected DL-alanine microfibers. In this work, the authors have reported the mechanism in forming the truss-like network of DL-alanine, the network structure, and microfibers topography. The manuscript has some areas to be improved. These include piezoelectric and mechanical characterizations as well as the demonstration of the material application. Overall, the manuscript is well-written and organized. The work is interesting and novel, yet there are several comments for the authors that might be useful to improve the quality of their studies:

A: We thank the reviewer for the high recognition of our work. These comments are very helpful for us to improve our manuscript.

1. The authors demonstrated that their strategy could be achieved on a variety of hydrophilic substrates. So, I wonder if the stretchability of the DL-alanine network depends on the growing substrate. It seems that the branching angles between DL-alanine microfibers (which determine the stretchability of the network) are different on different substrates (Figure S10). Please clarify it.

A: We thank the reviewer's insightful comment. The reviewer raised a very good point that the stretchability of the DL-alanine network may vary on dissimilar substrates because of the different branching angles. We agree that branching angles can affect stretchability in different directions by influencing the concentrated strains at the junction areas. Nevertheless, we cannot directly evaluate stretchability of the network on non-stretchable substrates. The growth substrate requires excellent stretchability so that the stretchability of the network can be evaluated. Except for PDMS, other substrates like metals, ceramics and polymers cannot withstand large tensile strains. Therefore, the stretchability of these systems is less meaningful to evaluate. This is the reason that we selected PDMS as the substrate and evaluated the stretchability of DL alanine/PDMS film as a whole. We added the discussion regarding the substrate effect on branching angle and stretchability in main text and highlighted it in yellow. (Page 9, *“Although different substrates may influence the stretchability of network itself by varying branching angles and density, the overall stretchability of a network-substrate system is only meaningful on stretchable substrates like PDMS.”*).

2. For the truss-like structure, it is recommended to the authors perform fatigue characterization, where they can apply cyclic strains to the network. This could be helpful in determining the functional lifetime of the network/device.

A: We thank the reviewer for this constructive suggestion. We performed fatigue test on the DL-alanine network on PDMS elastomer by applying cyclic strains. The DL-alanine/PDMS film was consistently stretched with a large strain (20%) and released to the original state for 100 cycles. The stress and strain of the film were recorded and the integrity of the DL-alanine network was also monitored during cycles. As shown in **Figure R1**, the hysteresis stress-strain loops do not show much deviation from each other under the 100 loading-unloading stretch cycles, which indicates a good stability of this DL-alanine network. The optical microscope indeed observed minor structural impairments at later stages. In the first 30 cycles, the DL-alanine is intact without

any observable changes. A very few dark-contrast spots were observed after the network was cycled 100 times, which might be due to the deviations from the original structure or limited detachments of MF from the substrate, but the majority of this network is complete. Thus, this network can currently withstand 100 cyclic loading-unloading tensile strains without significant structure failure. The stability and structural robustness may need to be further improved by optimizing the synthesis for practical application. We added the fatigue test (Figure R1) as Figure S15 in the Supplementary Information and added relevant discussion in the main text with highlights. (Page 10, “Fatigue test of DL-alanine network on PDMS was performed by applying cyclic strains. Only a very few dark-contrast areas appeared after 100 tensile straining cycles, suggesting the DL-alanine/PDMS film had a good stability with consistent stress-strain behavior over 100 large cyclic strains (Figure S15).”).

Figure R1 (Figure S15). a. Cyclic loading-unloading tensile straining of DL-alanine network on PDMS substrate. b. Optical microscopes of DL-alanine network after cycles of straining. The few dark-contrast spots in cycle 60 and 100 images indicate possible small deviations from the original structure.

3. The authors should investigate the interfacial interaction between the DL-alanine microfibers and the substrate. Will the network be buckling and detached from the substrate under bending?

A: We thank the reviewer for this good suggestion. The DL-alanine microfiber network has strong interaction with the PDMS substrate due to belt shape of individual microfiber and hydrophilicity of treated PDMS. We utilized a 3M VHB (very high bond) tape to demonstrate the strong adhesion, a commercial acrylic elastomer adhesive that was commonly used as a printing stamp. There are literatures showing that VHB can remove away nanomaterials/structures on PDMS substrates due to its excellent bonding strength and ease to process [1,2]. The VHB tape first conformally contacted the DL-alanine network on PDMS substrate, and then it was pulled away from the substrate with high peel velocity (10 cm s⁻¹), which is a sufficient speed leading to adhesion that is strong enough to adhere nano-materials/structures preferentially to the surface of the VHB [3]. As shown in **Figure R2**, unlike clean removal reported from weakly-adhered materials (e.g., percolated silver nanowires), there are still ~50% DL-alanine microfibers remaining on PDMS substrate, indicating the strong interfacial interaction between DL-alanine network and the PDMS substrate.

Figure R2 (Figure S16). Transferring DL-alanine network from PDMS substrate to commercial VHB tape. The left image is the DL-alanine fragments left on PDMS after transferring. The right image is the transferred DL-alanine fragments on VHB tape.

As for buckling, we did not observe apparent buckling during the entire tensile straining process (as shown in straining processes in both Figure S13 and S14). This could also be attributed to the strong adhesion of DL-alanine MFs to the substrate and good flexibility of the DL-alanine MFs. Detachments of the DL-alanine MFs could be observed after 45% tensile straining in transverse direction (Figure S13c). As shown in Figure R3a (Figure S13c), the thick black lines in the optical microscope image are air gaps caused by detached MFs (different refractive indexes between air and PDMS medium led to black boundary). Detachment can be also shown in SEM images (Figure R3b). We added the discussions on interfacial interaction and detachment of the MF under large strain in the main text and the Figure R2 and R3 as Figure S16 and Figure S13d and e in the Supplementary Information (Page 10, “Besides, the DL-alanine network had a strong interfacial interaction with the PDMS substrate, which was proved by the removal test using a 3M very high bond (VHB) tape (Figure S16). The strong interfacial bonding ensured consistent adhesion without observable buckling of MFs within the stretchable range. Small portions of detachment were only observed at exceeding large transverse tensile strains (>45%) (Figure S13d and e).”).

Figure R3 (Figure S13d and e). **a.** Detachment of DL-alanine MFs from substrate after 45% transverse strains. **b.** SEM image of a single DL-alanine MF detached from substrate.

Reference:

[1]. Zhang, Ziyang, et al. "Stretchable transparent wireless charging coil fabricated by negative transfer printing." *ACS Applied Materials & Interfaces* 11.43 (2019): 40677-40684.

[2]. Liang, Jiajie, et al. "Elastomeric polymer light-emitting devices and displays." *Nature Photonics* 7.10 (2013): 817-824.

[3]. Meitl, Matthew A., et al. "Transfer printing by kinetic control of adhesion to an elastomeric stamp." *Nature Materials* 5.1 (2006): 33-38.

4. Although DL-alanine has been reported to exhibit piezoelectricity, it is still recommended to have non-piezoelectric control samples (e.g. PDMS) when the author characterizes the piezoelectric coefficient. As the piezo-coefficient reported in the manuscript is relatively small (around 6 pC/N), I am concerned it may be interfering with triboelectricity.

A: We thank the reviewer for the thoughtful suggestion. We agree that the triboelectric effect may contribute to the piezoelectric output, when there are small contact and separation actions of two dissimilar materials during the measurement. We are fully aware of this potential artifact and the measurement was carefully designed to minimize the contact and separation between dielectric materials when applying tapping force for piezoelectricity measurements. The 3 N tapping force was applied by pressing the PDMS substrate with another PDMS rod and there was no separation between PDMS materials during cyclic forces (schematic of setup in Figure R4a). Therefore, the triboelectric effect was effectively minimized. For confirmation, we test the PDMS substrate without DL-alanine network. There were no regular signals in both voltage output and current output, as presented in Figure R4b. We emphasized the negligible triboelectric contribution in the main text and added Figure R4 as Figure S19 in the supporting information. (Page 13: "A 3 N tapping force was applied by pressing the PDMS substrate with another PDMS rod. The PDMS rod remained a close contact with the PDMS substrate during the cyclic pressing to minimize the triboelectric contribution (Figure S19). The near zero outputs of pure PDMS substrate without DL-alanine network further confirmed the negligible triboelectric contributions to the piezoelectricity measurements.")

Figure R4 (Figure S19). a. Schematic of experiment set-up for piezoelectric coefficient measurement. b. Voltage and current output of pure PDMS substrate without DL-alanine network.

5. In addition, the authors should provide more data regarding its piezoelectric performance. For instance, output voltages under different applied strains/forces or the polarity experiment (swapping electrodes).

A: We thank the reviewer for the suggestion. A series of strains (0%, 5%, 10%, 20%) and tapping forces (1N, 3N, 6N) have been applied to the DL-alanine network. The corresponding voltage outputs are recorded and displayed in Figure R5. As shown, constant tapping force under different strains lead to peak-to-peak voltage output of ~ 90 mV without significant variations (Figure R5a). On the contrary, the voltage output increases significantly from ~ 40 mV to ~ 150 mV when larger tapping force is applied. (Figure R5b). Forward and reverse connections between electrodes and electrometer probes resulted in voltage outputs with opposite polarity (Figure R5c), indicating the real piezoelectric signals instead of artifacts. We added the relevant discussion in the main text and added Figure R5 as Figure S20 in the supporting information. (Page 13: “Constant 3N tapping force under different strains led to peak-to-peak voltage output of ~ 90 mV without significant variations, whereas the voltage output increased significantly from ~ 30 mV (1 N) to ~ 150 mV (6 N) when larger tapping forces were applied (Figure S20).” Supplementary Information, Page 23: “Forward and reverse connections between electrodes and electrometer probes resulted in voltage outputs with opposite polarity, indicating real piezoelectric signals.”).

Figure R5 (Figure S20). a. Voltage output of DL-alanine piezoelectric network under a series of strains (0%, 5%, 10%, 20%). b. Voltage output of DL-alanine piezoelectric network under different forces (1-6 N). c. Voltage output of DL-alanine piezoelectric network under forward and reverse connections between electrodes and electrometer probes.

6. Please justify what is the piezoelectric coefficient (e.g., d_{31} , d_{33} , or d_{15}) that the authors measured.

A: We thank the reviewer's comment. We measured effective piezoelectric coefficient d_{13}^{eff} of the whole network in Figure 3d. This d_{13}^{eff} is different from the intrinsic piezoelectric coefficient of DL alanine MFs. The definition of d_{13}^{eff} follows the standard definition of piezoelectricity:

$$P_i = d_{ij} \sigma_j \quad (i = 1, 2, 3; j = 1, 2, \dots, 6)$$

where P is the polarization, d is the piezoelectric coefficient and σ is the stress. As illustrated in Figure R6, the out-of-plane direction is "3" direction of DL alanine/PDMS film, where the in-plane longitudinal direction (growth direction) and transverse direction are "1" direction and "2" direction, respectively. Since the parallel electrodes were placed to collect the longitudinal polarization built on the "1" direction of network and the tapping force is applied along its "3" direction, the effective piezoelectric coefficient we measured is the d_{13}^{eff} of the entire DL alanine network film.

Figure R6 (Revised Figure 3d). Schematics of DL-alanine network and electrode configuration for measuring piezoelectric coefficient. Axes indicate the effective directions.

In addition, we also measured d_{33} of the DL-alanine MFs. Please see the detailed responses in the next comment.

We revised Figure 3d based on Figure R6, added the description of effective piezoelectric coefficient in the main text and added its discussion in the Supplementary Information as Supplementary Methods (Page 13: *"The effective piezoelectric coefficient d_{13}^{eff} of the entire DL-alanine network film is defined as the in-plane polarization resulted from vertical displacement (Supplementary Methods)."*).

7. Can the authors perform PFM to assess the piezo coefficients of single microfiber? Do the branding angles and orientations of the microfibers in the networks affect the overall effective piezoelectric performance of the film?

A: We thank the reviewer for the questions. To assess the piezoelectric coefficients of single microfiber, we performed vertical PFM test under different driving voltages on a single microfiber. As presented in Figure R7, the DL alanine microfiber exhibited strong vertical responses. Uniformly strong phase responses in vertical direction indicated high out-of-plane polarizations whereas piezoelectric coefficient d_{33}^{eff} can be assessed by amplitude responses under different

driving voltages. Calibrated by standard LiNbO₃ sample, this DL alanine revealed a longitudinal piezoefficient d_{33}^{eff} of ~5.5 pm.

Figure R7 (Figure S18). a. Vertical PFM phase response of single DL-alanine MF. b. Vertical PFM amplitude response of single DL-alanine MF. c. PFM amplitude responses of single DL-alanine under different driving voltages.

The reviewer raised a very good point in piezoelectric performances of DL alanine network with varying branching angles and MFs orientations. Currently, we cannot selectively control the branching angles and orientations of DL alanine microfibers during the growth. However, the influences of branching angles and orientations can be anticipated by analyzing the in-plane and out-of-plane polarizations. Larger branching angles and misorientations might result in less net in-plane polarization along the growth direction, weakening in-plane piezoelectric responses of the DL-alanine network. However, because out-of-plane polarization is only dependent on how the out-of-plane dipole is oriented, the out-of-plane piezoelectric response of DL-alanine network is independent of branching angle and orientations.

We added the effective piezoelectric coefficient of single microfiber in the main text and inserted Figure R7 as Figure S18 in the Supplementary Information (Page 12: “The piezoelectric coefficient, d_{33}^{eff} of single DL-alanine MFs, can be assessed by amplitude responses under different driving voltages (Figure S18). Calibrated by standard LiNbO₃ sample, this DL alanine MF revealed a d_{33}^{eff} of ~5.5 pm.). We also added the discussion of the influence of branching angles and orientations in the main text (Page 14: “Several factors can affect the piezoelectric responses of the network. Larger branching angles and misorientations might result in less net in-plane polarization along the growth direction, resulted in variations of in-plane piezoelectric responses of the DL-alanine network. However, the branching and orientation should not affect the net out-of-plane polarization and the out-of-plane piezoelectric responses of the network.”).

8. The advantage of DL-alanine is its biocompatibility and mostly biodegradability, which makes it favorable for implantable applications. Yet, the authors utilized their material for a wearable tactile sensor, which diminishes the benefits of the materials, especially since there have been extensive reports of other piezoelectric materials (e.g., PVDF, PLLA, PZT composites, etc.) for wearable sensors with higher performance. The work could be more significant if the authors could grow the DL-alanine networks on a flexible biodegradable substrate (instead of growing on PDMS) to make the device completely biodegradable and use it to detect the leg movements in swine.

A: We thank the reviewer for this insightful suggestion. We agree with the reviewer that DL alanine is an ideal building block for transient implantable devices due to its biocompatibility and biodegradability. Although making DL alanine-based transient implantable device has high significance, our current work is focused on the omnidirectional stretchability and conformability of a novel DL alanine network through self-assembly. Other piezoelectric materials such as PVDF[1], PLLA[2], and PZT composites[3] indeed have been applied as wearable sensors, and biodegradable implantable devices made of PLLA has also been demonstrated [2,4]. However, tissue-comparable omnidirectional stretchability together with conformability have not been realized in these wearable sensors. Complicated fabrications (e.g., 3D printing and lithographic patterning), which may compromise piezoelectric performance, only enable stretchability in one direction, mismatching the mechanical behavior to tissues. This can potentially lead to device detachment, scar formation and even failure. The significance and focus of this work are therefore to demonstrate a self-assembled piezoelectric DL-alanine crystal network with omnidirectional stretchability, which has the potential for making a fully stretchable and conformal implantable piezoelectric device seamlessly interfacing tissues and withstanding multidirectional strains in vivo.

Indeed, we also demonstrated the growth of our DL alanine microfiber network on a flat biodegradable substrate in Figure S10d (Supporting Information). We utilized a polycaprolactone (PCL) material known for softness and biodegradability as substrate. Nevertheless, the PCL is not an elastomer, which does not have stretchability. The device made on PCL cannot work under strain. Besides, how to make stretchable and biodegradable electrodes is another big issue that needs to be addressed in order to reach a fully biodegradable and stretchable device. These are beyond the scope of this work. We added all relevant discussions on making a biodegradable, implantable, and stretchable piezoelectric DL-alanine device in the main text and highlighted them. (Page 17, *“Although other piezoelectric materials such as PVDF, PLLA, and PZT composites have been demonstrated as wearable and implantable sensors, this work distinguishes from them by enabling a self-assembled piezoelectric DL-alanine crystal network with omnidirectional stretchability, which provides a critical component for achieving fully stretchable and conformal implantable piezoelectric devices. Future research should enable similar biodegradability and omnidirectional stretchability in electrodes and substrates in order to realize the envisioned devices.”*).

References:

[1] Shepelin, Nick A., et al. "New developments in composites, copolymer technologies and processing techniques for flexible fluoropolymer piezoelectric generators for efficient energy harvesting." *Energy & Environmental Science* 12.4 (2019): 1143-1176.

[2] Curry, Eli J., et al. "Biodegradable piezoelectric force sensor." *Proceedings of the National Academy of Sciences* 115.5 (2018): 909-914.

[3] Hong, Ying, et al. "Highly anisotropic and flexible piezoceramic kirigami for preventing joint disorders." *Science Advances* 7.11 (2021): eabf0795.

[4] Curry, Eli J., et al. "Biodegradable nanofiber-based piezoelectric transducer." *Proceedings of the National Academy of Sciences* 117.1 (2020): 214-220.

Reviewer #2 (Remarks to the Author):

This manuscript describes self-assembly of DL-alanine possessing truss-like microfiber structures, its piezoelectric and stretchability characterization, and their application on the wearable/implantable setting. Stretchability is a key property for wearable devices to adapt our bodily movement. Piezoelectricity has broad application potentials as a wearable device. However, most of inorganic piezoelectric materials are crystal based so that brittle or not stretchable, and not-biocompatible. In this manuscript, the authors synthesized organic microfiber structure using DL-alanine. The resulting microfiber structures with truss-like network structures can adapt mechanical stretching in an omni-directional manner. They also fabricate stretchable device using PDMS/Ag nanowire substrates and demonstrated their biocompatible and motion sensing properties. The result is interesting and shows the full spectrum of piezoelectric materials from synthesis, characterization, and in vivo demonstration. However, although the stretchability part might be novel, many components of the observation or characterization are not that novel. In addition, it is limited to attract broad interest. Therefore, this reviewer thinks it is not suitable for *Nature Comm*. Instead, it might be better suit for more specific journal specialized for nanoscience, energy, or relevant field.

We thank the reviewer for detailed comments and efforts towards improving our manuscript. We appreciate the reviewer's positive opinion on the novelty of stretchability in our work. We want to argue that this key novelty will bring broad and substantial impacts to many research disciplines, such as materials science (e.g., biomaterials), nanotechnology (e.g., nanomaterials self-assembly) and wearable and implantable devices. This work enables a critical component for realizing *fully stretchable and conformal implantable piezoelectric devices*.

Here are more specific points that need to be addressed further.

- Many amino acids exhibit piezoelectric properties. In this work, the authors used a racemic mixture of D- and L-alanine. The authors need to explain why DL-alanine and how this racemic mixture affect the polarization of the materials. Most of the schematic illustration in Fig 1g and 2f does not indicate or reflect the racemic nature of the materials. Furthermore, does D-alanine, or L-alanine have the same self-assembling properties or exhibit mirror symmetric results in piezoelectric? How racemic mixture of the compound can generate polarized structures and exhibit SFG in Fig

A: We thank the reviewer's question. We agree with this reviewer on more explanations of racemic DL alanine and its superiority over L and D enantiomers. Indeed, DL alanine is one of the strongest

piezoelectric materials among common amino acids and their enantiomers[1-3]. The racemic alanine amino acid has different crystal structure compared to D and L counterparts, which results in significantly stronger piezoelectric behavior.

As illustrated in Figure R8, both D and L alanine crystalize with orthorhombic crystal symmetry and a space group of $P2_12_12_1$. Although noncentrosymmetric, D alanine and L alanine molecules pack as antiparallel layers. As a result, their molecular dipoles in the unit cell (indicated by the green arrows) cancel out by each other in both longitudinal and transverse directions. D alanine and L alanine crystals thus only have weak shear piezoelectric responses [3]. On the contrary, DL alanine crystalizes with a different space group of $Pna2_1$ (as mentioned in the manuscript). Its molecular dipoles are well aligned in both longitudinal and transverse directions, giving high piezoelectric coefficients correspondingly. As for the schematic illustrations in Figure 1g and 2f, the racemic nature of the materials is indeed included. For reviewer's convenience, the L alanine molecules and D alanine molecules in the schematic structures are identified and pointed out by red arrows, as presented in Figure R9 below.

Figure R8. Schematics of D and L alanine molecules (top panel) and crystal structures of L alanine, D alanine and DL alanine (bottom panel).

Figure R9. a. Schematic illustration in Figure 1g. **b.** Schematic illustration in Figure 2f. D and L enantiomers are pointed out by red arrows.

The second harmonic generation (SHG) is a non-linear optical property owned by noncentrosymmetric media. As illustrated in Supplementary Text of Supporting Information, nonlinear polarization in SHG of piezoelectric crystals can be described by the following equation:

$$P_i(2\omega) = 2\varepsilon_0 \sum_{jk} d_{ijk} E_j(\omega) E_k(\omega)$$

where ε_0 is the permittivity of free space, d is the second-order non-linear coefficient, and E is the electric field in electromagnetic waves. The symmetry requirements for both SHG and piezoelectricity are identical, as both are described mathematically by the third rank tensor d_{ijk} . Since DL alanine has nonzero and large piezoelectric coefficients, it has strong SHG signals correspondingly.

To highlight the significance of DL alanine and its superior piezoelectricity, we added relevant descriptions in the introduction of this manuscript. (Page 4, "*DL alanine is one of the strongest piezoelectric amino acids. Unlike the D or L alanine that only have weak shear piezoelectricity, DL alanine crystalizes with its molecular dipoles well aligned in both longitudinal and transverse directions, leading to high piezoelectric coefficients.*"). We also revised Figure 1g and 2f, with D and L alanine molecules pointed out by red arrows.

References

- [1] Lemanov, V. V. "Ferroelectric and piezoelectric properties of protein amino acids and their compounds." *Physics of the Solid State* 54 (2012): 1841-1842.
- [2] Guerin, Sarah, Syed AM Tofail, and Damien Thompson. "Organic piezoelectric materials: milestones and potential." *NPG Asia Materials* 11.1 (2019): 10.
- [3] Guerin, Sarah, et al. "Racemic amino acid piezoelectric transducer." *Physical review letters* 122.4 (2019): 047701.

- In Fig 1f, need more description to explain what is 2D XRD and how this 2D XRD experiment was performed. I think the 1f left, right is mis-labelled between up and down.

A: We thank reviewer's comments. 2D XRD is two-dimensional X-Ray diffraction. Unlike traditional XRD, which only provides a one-dimensional diffraction pattern, 2D XRD provides a two-dimensional image that can reveal more comprehensive information about the crystal structure (e.g., orientation and texture). In 2D XRD, the whole or a large portion of the diffraction rings can be measured simultaneously. The radius and brightness of the diffraction rings represent scattering angles and intensity of the X-ray peaks. In our work, the 2D XRD was performed in Bruker D8 Discovery facility (Methods), and DL-alanine powders and MF networks were scanned under Cu K α radiation. The reviewer is correct that the labels in figure 1f are wrong. We corrected the label as up and down (Page 25, Figure 1f caption.). We defined 2D XRD and added relevant descriptions in its first appearance and corrected the label error in figure caption. (Page 6, "*The crystallographic alignment of the MF network was further validated by the two-dimensional (2D) XRD (Figure 1f). The 2D XRD image can reveal more detailed information about the crystal*")

structure, such as orientation and texture. Crystal orientation can be indicated by the pattern of the ring.).

- Fig s4 schematic: (410), (210) (110) was specified on the one facet. Need more explanation. It might need more data to link the AFM image with 410, 210, 110.

A: We thank the reviewer for the comment. The schematic in Figure S4a is just an illustration indicating side facets, e.g. (410), (210) and (110). They are not necessarily exposed simultaneously. It is common that many MFs have only one crystal facet on their sides (Figure S4b). To provide more supports, we calculated the angles between lateral facets and top/bottom facets of our MFs based on the height scans of AFM and compared them with theoretical angles between different crystal facets. These results matched well.

According to the crystal information of DL-alanine (PDF#21-1569), the theoretical angles between (410) and (010), (210) and (010), (110) and (010) are 63.51° , 45.09° , and 26.64° , respectively. The angles between lateral side and bottom side based on AFM height scans in Figure S4 are calculated and displayed in Figure R10. There are two types of facets in both left and right lateral sides of DL-alanine in Figure R10a. The angles between these facets and the bottom side are calculated as 23.3° and 45.6° based on coordinates of turning points, which are very close to 26.64° ((210) and (010)) and 45.09° ((110) and (010)) given the measurement errors. Figure R10b shows another individual MF having only one facet in each lateral side. The measured angle on the left side is 30.2° , matching the value between (110) and (010). The angle on the right side (47.4°) is very close to the value between (210) and (010). Therefore, the left side is (110) while the right side is (210). Although (410) was not observed in these two MFs, it could be found on other MFs (Figure R10c). Our assumptions that the lateral side of DL alanine can either expose multiple facets (Figure S4a) or one type of facet (Figure S4b) among (410), (210) and (110) are validated by measuring the interfacial angles.

Figure R10 (Revised Figure S4). a. AFM and height profile of a single DL alanine MF with both lateral sides consisting of (110) and (210) facets. b. AFM and height profile of a single DL alanine MF with left lateral side consisting of (110) facet while right lateral side consisting of

(210) facet. c. AFM and height profile of a single DL alanine MF with both lateral sides consisting of (210) and (410) facets.

We reorganized Figure S4 with enlarged height profiles together with calculated angles and added the discussions in the main text. (Page 7, “The (110), (210) or (410) side facets could be verified by measuring the angles between these facets and top/bottom side of MFs and comparing them with theoretical angles between the crystal facets.”) and Figure S4 captions in Supporting Information. (Page 7 Supporting information, “According to the crystal information of DL-alanine (PDF#21-1569), the theoretical angles between (410) and (010), (210) and (010), (110) and (010) are 63.51°, 45.09°, and 26.64°, respectively. The real angles between different facets can be calculated based on coordinates of turning points in AFM height profiles of single MFs.”)

- Fig s6: Contact angle measurement is confusing. By mixing EtOH: H2O in different ratio, it does not exhibit the consistent contact angle. Need more explanation. Typo? (tensor => tension?)

- Each micrograph needs to indicate where film growth (puling) direction is.

A: We thank the reviewer for the comment. This non-linear change of contact angles under different EtOH/H₂O ratios is mainly due to the huge difference of surface tension between water and ethanol. It is known that water has a rather high surface tension around 72.75 mN/m while ethanol has a low surface tension of only 22.31mN/m.[1] The surface tension of water-ethanol mixture does not show a complete linear change as the volume percentage of ethanol increases. At low percentage, adding ethanol can dramatically lower the surface tension of the mixture solution. Therefore, the contact angle drops quickly from pure DI water to 1:3 (water: ethanol) mixture solution, and to 1:6 mixture solution. Once the ethanol has a volume percentage over 85% (1:6), the surface tension of mixture reaches 24.32 mN/m [2], which is close to the value of pure ethanol. The mixture solution can completely wet the solid surface as pure ethanol and there is no distinguishable change of contact angle by further increasing ethanol. We added these explanations and discussions in the main text. (Page 8, “The non-linear change of contact angles under different ethanol/water ratios is mainly due to the huge difference of surface tension between water and ethanol. Once the ethanol to water ratio is over 6, the surface tension of the solution is very close to pure ethanol and can completely wet the DL-alanine MF”).

We also thank the reviewer for going through the details. We corrected the typo “tensor” as “tension” and added labels and the pulling direction the first image in Revised Figure S6. For reviewer’s convenience, Revised Figure S6 was presented as Figure R11 as below.

Figure R11 (Revised Figure S6). Contact angles of different solutions on DL-alanine micro-crystals by optical microscopes. The solutions are water and biphasic solutions consisting of water and ethanol with varying compositions. The difference in contact angles is associated with different surface tension that deflects of attachment of nanofibrils. The pulling direction, DL-alanine MF and solution are indicated by red arrows.

References:

[1] Vazquez, Gonzalo, Estrella Alvarez, and Jose M. Navaza. "Surface tension of alcohol water+ water from 20 to 50. degree. C." *Journal of chemical and engineering data* 40.3 (1995): 611-614.
[2] Voloshin, Galina. *Cellulose Nanocrystal Aqueous Inks Evaluated for Printed Electronics and Application to Thin-film Transistors*. MS thesis. University of Waterloo, 2016.

- Fig S9E: Inset height profile need more explanation. Redline seems not match with the profile.

A: We thank the reviewer for the comment. The height profile is indeed obtained by performing an AFM scan on the same trench presented in Figure S9E. While roughness may vary in the same trench, local details can hardly be revealed in this low-magnification SEM. To avoid confusion, we removed the redline and the height profile in the SEM image (Figure R12a). The height profile of the trench together with the original AFM image is added as Figure S9F (Figure R12b). For the reviewer's convenience, the newly added Figure S9F is presented below. This figure has been added to the Supporting Information. (Supporting Information, Page 12).

Figure R12. a (Revised Figure S9E). SEM images of DL-alanine MFs crossing over a trench (Figure S9F). Local AFM image of the trench and height profile of the trench as marked by the red line.

- Is it possible to characterize d_{33} or d_{14} values from the observed microfibers because it is large enough?

A: We thank the reviewer for this question. Longitudinal piezo-coefficient d_{33} of the DL alanine microfibers can be estimated by vertical PFM characterization. To assess it, we performed vertical PFM test under different driving voltages on a single microfiber by utilizing a Bruker's Dimension Icon AFM. As presented in Figure R13, the DL alanine microfiber exhibited strong vertical responses. Uniformly strong phase responses in vertical direction indicated high out-of-plane

polarizations whereas piezoelectric coefficients d_{33}^{eff} can be assessed by amplitude responses under different driving voltages. Calibrated by standard LiNbO₃ sample, this DL alanine revealed a longitudinal piezocoefficient d_{33}^{eff} of ~5.5 pm. Nevertheless, transverse piezo-coefficient (e.g., d_{31}) and shear coefficient (e.g., d_{15}) can hardly be assessed as they are usually coupled in lateral PFM test due to the complex torsional deflection of the AFM tip. We added the discussions of piezoelectric coefficient in the main text with highlight and Figure R13 as Figure S18 in the Supplementary Information (Page 12, “The piezoelectric coefficient of single DL-alanine MF can be assessed by amplitude responses under different driving voltages. Calibrated by standard LiNbO₃ sample, this DL alanine revealed a longitudinal piezocoefficient d_{33}^{eff} of ~5.5 pm”).

Figure R13 (Figure S18). a. Vertical PFM phase response of single DL-alanine MF. b. Vertical PFM amplitude response of single DL-alanine MF. c. PFM amplitude responses of single DL-alanine under different driving voltages.

- Fig S17: There is a schematic to indicate the charge on +/- . Based on the described structures in AFM images on Fig S4, schematic needs more description how the polarized + and – surface exist.

A: We thank the reviewer for the comment. The existence of polarized + and – surface is due to the induced out-of-plane dipoles under stress/strain. We apologize for the misplacement of charges in Figure S17 that may cause confusion. We redraw the schematic by simplifying the charges and connecting it with MF structure. As measured by the PFM (Figure R13), the MF showed high out-of-plane responses, which indicates strong out-of-plane dipoles. Meanwhile, the cross-section SEM image in Figure S17b revealed that Ag nanowires (NWs) closely contact with the lateral and top sides of the DL alanine MFs. Without stress, no charges due to piezoelectricity built up on the interface (Figure R14-i). When stress is applied, the out-of-plane deformation can induce out-of-plane dipoles, and positive charges are accumulated on the lateral and top sides of

MFs. Negative charges would be drawn from the ground into Ag NWs electrode to screen and stabilize these positive charges on the MFs. (Figure R14-ii). Once the stress is released, the induced dipole with accumulated surface charges disappear in MFs. As a result, the negative charge on the Ag NWs would flow back to the ground (Figure R14-iii). We replaced the previous Figure S17a with Figure R14 and added more descriptions in the figure caption (Supplementary Information, Page 20: “a. Schematic of working mechanism of the single-electrode stretchable piezoelectric NG. i. Without stress, the charges between MF and Ag electrodes are balanced. ii. When stress is applied, the out-of-plane dipoles and accumulated positive charges on the lateral and top side of MFs draw negative charges from the ground into Ag NWs electrode. iii. Once the stress is released, the induced dipole and surface charges disappear in MFs, which drive the negative charge on the Ag NWs to flow back to the ground.”).

Figure R14 (Revised Figure S17a). Schematic of the working mechanism of the single-electrode stretchable piezoelectric NG. i. Without stress, no charges due to piezoelectricity built up on the interface. ii. When stress is applied, positive charges accumulated on the lateral and top side of MFs draw negative charges from the ground into Ag NWs electrode. iii. Once the stress is released, the induced dipole and surface charges disappear in MFs, which drive the negative charge on the Ag NWs to flow back to the ground.

Reviewer #3 (Remarks to the Author):

The authors present a study regarding the fabrication and piezoelectric properties of peptide-based networks exhibiting high degree of stretchability, while maintaining piezoelectricity, thus allowing conformable and stretchable piezoelectric devices.

I think the paper is very well written, the procedures and results are clearly described and presented. I have a few comments for the authors to address which I believe would improve the related discussion and the readability of the paper:

A: We very much appreciate the reviewer's high recognition of our work. We really thank the thoughtful comments and efforts towards improving our manuscript.

1. It appears that the main novelty is in the controlled fabrication of the peptide truss and the ability to use it as an electromechanical device. If that is the case, a clearer comparison to some previous work on conformal PE devices is missing (some is already cited), in terms of efficiency, ease of production etc'.

A: We thank the reviewer for this constructive comment. We agree that a comparison to some previous works can better present our novelty and significance. We compiled the information of some representative conformal piezoelectric device in the table, in terms of outputs, force, production method, and stretchability. The table is presented as Table R1 as below. As most conformal piezoelectric devices are applied as sensors instead of energy harvesters, the efficiency is less provided in these works. Our calculated energy efficiency is approximately 1.08%, which is comparable to the energy efficiency (1.76%) of previously reported conformal piezoelectric devices. Compared to other works, our major advantages are omnidirectional stretchability and conformability together with easy synthesis based on self-assembly.

Table R1. Comparison of Conformal Piezoelectric Device

Electromechanical Materials	Outputs /Force	Fabrication	Stretchability	Application	Efficiency	Ref
DL-alanine network	0.1 V/3 N	Self-assembly	20-40% all direction	In vivo sensor	1.08 %	this work
PZT kirigami	1 V/30 kPa	Template assisted sol-gel/poling	100% single direction	Joint sensor	-	8
P(VDF-TrFE)	0.7 V/-	Lithography/spin-coating	30% single direction	Energy harvester	-	9
BaTiO ₃ -P(VDF-TrFE)	6 V/60 N	3D printing	300% single direction	Gait sensor	-	10
PZT ribbons	4 V/100 N	Lithographic patterning	Not stretchable	In vivo energy harvester	1.77%	[1] (41)

ALN thin film	0.06 V/6 N	Lithographic patterning/Sputtering	Not stretchable	Facial Sensor	-	[2] (42)
ALN (piezo)/PDMS (tribo)	8 V/5 N	Sputtering/UV patterning/Film assembly	120% single direction	Wearable Sensor	-	[3] (43)
Boron nitride nanocomposite	5 V/40 N	3D printing/poling	Not stretchable	Robotic sensor	-	[4] (44)

We added the discussion on comparison to previous works in main text and cited more related papers. (Page 5, “Compared to other works on conformal piezoelectric devices (Table S1)^{8-10,41-44}, our major advantages are omnidirectional stretchability and conformability together with easy synthesis based on self-assembly.”). We also added Table R1 as Table S1 in the Supplementary Information with highlight.

References:

- [1] Dagdeviren, Canan, et al. "Conformal piezoelectric energy harvesting and storage from motions of the heart, lung, and diaphragm." *Proceedings of the National Academy of Sciences* 111.5 (2014): 1927-1932.
- [2] Sun, Tao, et al. "Decoding of facial strains via conformable piezoelectric interfaces." *Nature biomedical engineering* 4.10 (2020): 954-972.
- [3] Mariello, Massimo, et al. "Conformal, Ultra-thin Skin-Contact-Actuated Hybrid Piezo/Triboelectric Wearable Sensor Based on AlN and Parylene-Encapsulated Elastomeric Blend." *Advanced Functional Materials* 31.27 (2021): 2101047.
- [4] Zhang, Jie, et al. "3D printed piezoelectric BNNTs nanocomposites with tunable interface and microarchitectures for self-powered conformal sensors." *Nano Energy* 77 (2020): 105300.

2. Some Figures involve several experiments; I believe it can be confusing and the paper can benefit from their separation.

A: We thank the reviewer for the suggestion. We appreciate this suggestion to improve the readability of our manuscript. Admittedly, many figures in this manuscript group more than one experiment. However, each figure is designed to include experiments that deliver related information on the same topic and complement each other. Specifically, multiple characterizations (such as AFM, SEM, XRD, etc.) in Figure 1 all describe the topography/morphology of the DL alanine network. This helps present growth mechanism of truss-like structure and enhance the clarity. All optical and mechanical characterizations and simulations in Figure 2 aim to demonstrate the omnidirectional stretchability of DL alanine network. Unifying them together can better highlight the relationships between geometry/structure and stretchability and make the stretching results more convincing. Although Figure 3 involves piezoelectric and biological characterizations, these two intrinsic properties of DL alanine complement each, and together allowing for the application of DL-alanine network in wearable and implantable devices. Lastly, Figure 4 connects all experiments/characterizations at the device level for easy comparison. Again, we agree with the reviewer that involving multiple unrelated experiments in a single figure may lead to confusion. But here, combining correlated sub-figures into a single figure in this manuscript allows us to present our work in a more logic and efficient manner.

3. Regarding the three-point bending type current generation (Fig 3). The experimental setup is not well described in the main text. I suggest improving the schematic. Was the measurement always done in the direction of the fibers? It is not clear. In that regard - what were the conditions of current recording? For low current sensitivities (as were probably used), a high input resistor affects the current and therefore the direct inference of effective d coefficient is not straightforward (it is no longer short-circuit current). I urge the authors to examine the conditions used and analyze the experimental data accordingly.

A: We thank the reviewer for the insightful comments. We agree that more descriptions of experimental setup for Figure 3d is necessary. The current measurement of DL-alanine network is not under a three-point bend test. In fact, the DL-alanine/PDMS film is first fixed on a pair of stages and a tensile strain of 20% was applied by moving apart the stages. Electrodes directly contacting DL alanine network were then applied together with another pair of stages for fastening. A tapping force of $\sim 3\text{N}$ at 1 Hz was then applied for testing the piezoelectric responses under strains. To make the measurement consistent and comparable, the electrodes were always applied perpendicular to the growth direction (the direction of fibers). The schematic was presented as Figure R15 below.

Figure R15. a (Figure S19a). Schematics of experimental set-ups for evaluating piezoelectric coefficient. b (Revised Figure 3d). Schematics of DL-alanine network and electrode configuration during piezoelectric coefficient measurement. Axes indicate the effective directions.

The reviewer is correct regarding low current measurement. When measuring weak current, a high input impedance was usually utilized for amplifying the signals, which may lead to the distortion of original current inputs. Here, we used a Stanford SR570 low-noise preamplifier (Materials and Methods in Supporting Information). As a pre-amplifier typically has a high input impedance and a low output impedance, it allows for amplifying weak signals without introducing noise or distortion. Previously, many works have utilized preamplifier for measuring low current outputs from piezoelectric materials and estimated the piezoelectric coefficient.[1-3] Meanwhile, to maximize the accuracy of measurement, low noise mode, low input offset current and low-pass filter in the pre-amplifier are all adapted in our test to remove noise from the measurement signal. Furthermore, a customized metal box as shielding is harnessed. This can be particularly effective in reducing distortion caused by external sources. By minimizing noise and distortion, we think the accuracy of the current measurement can be maximized, leading to reliable estimation of effective d coefficient.

We revised the schematic in Figure 3d by adding electrodes (main text, Page 28). We added another schematic of test set-up as Figure S19a in the Supplementary Information

(Supplementary Information, Page 22). The detailed description of current measurement is also added in the Methods in main text (Page 19, *“To maximize the accuracy of measurement, low noise mode, low input offset current and low-pass filter in the pre-amplifier are adapted to remove noise from the measurement signal. Furthermore, a customized metal box as shielding is harnessed which can reduce distortion caused by external sources”*), together with the discussions on the accuracy added in the main text (Page 13: *“High accuracy of piezoelectric coefficient estimation based on low current measurement by preamplifier can be achieved by optimizing the test conditions (Methods), which have been demonstrated by previous works^{54,55.}”*).

References:

- [1] Zhong, Junwen, et al. "A flexible piezoelectret actuator/sensor patch for mechanical human-machine interfaces." ACS Nano 13.6 (2019): 7107-7116.
- [2] Zhang, Zhuolei, et al. "Tunable electroresistance and electro-optic effects of transparent molecular ferroelectrics." Science Advances 3.8 (2017): e1701008.

4. Which d coefficient is dominant in the measurements? Is it d₃₃? How do the calculated values (or corrected ones) compare to the expected d₃₃? In fact, I'm not sure the paper suggests a value for the piezoelectric coefficients.

A: We thank the reviewer's comment. In this work, we measured both the effective piezoelectric coefficient d₁₃ of the whole network and the effective piezoelectric coefficient d₃₃ of individual DL-alanine MF (Please see the answer to Question 5) by different approaches. While the effective d₁₃ of network is estimated based on the setup in Figure 3d, effective d₃₃ of single DL-alanine is evaluated by PFM.

The measurement of d_{13}^{eff} is because of the definition of piezoelectricity:

$$P_i = d_{ij} \sigma_j \quad (i = 1, 2, 3; j = 1, 2, \dots, 6)$$

where P is the polarization, d is the piezoelectric coefficient and σ is the stress. As illustrated in Figure R15b, the out-of-plane direction is “3” direction of DL alanine/PDMS film, where the in-plane longitudinal direction (growth direction) and transverse direction are “1” direction and “2” direction, respectively. Since the parallel electrodes were placed to collect the longitudinal polarization built on the “1” direction of network and the tapping force is applied along its “3” direction, the effective piezoelectric coefficient we measured is the d_{13}^{eff} of the whole DL alanine network. The piezo-responses during d_{13}^{eff} measurement is mostly contributed by in-plane polarization of DL alanine MFs.

The measurement of d_{33}^{eff} is based on converse piezoelectric effect:

$$S_j = d_{ij} E_i \quad (i = 1, 2, 3; j = 1, 2, \dots, 6)$$

where S is the strain, d is the piezoelectric coefficient and E is the electric field. During the vertical PFM measurement, the electric field is applied between the tip and the substrate along “3” direction (out-of-plane direction) of DL-alanine MF, and the vertical displacement of DL-alanine MF along “3” direction is measured by the PFM tip. As a result, the effective piezoelectric

coefficient we measured is the d_{33}^{eff} of the DL-alanine MF. The piezo-response during vertical PFM measurement is mostly contributed by out-of-plane polarization of single DL alanine MF.

As for the comparison to expected piezoelectric coefficient, some previous studies have indicated that DL alanine crystals has piezoelectric coefficient in the range of 1-10 pC/N [1,2]. Our values of 4-6 pC/N are consistent with these results. We added the description of effective piezoelectric coefficient in the main text and added its discussion in the Supplementary Methods of Supplementary information (Page 13: “*The effective piezoelectric coefficient d_{13}^{eff} of the entire DL-alanine network film is defined as the in-plane polarization resulted from vertical displacement (Supplementary Methods).*”. Page 13: “*The estimated values of 4-6 pC/N are consistent with piezocoefficient results of DL-alanine crystals from previous studies^{39,56}.*”).

References:

[1] Guerin, Sarah, et al. "Racemic amino acid piezoelectric transducer." *Physical review letters* 122.4 (2019): 047701

[2] Jeon, Buil, Dongsoo Han, and Giwan Yoon. "Piezoelectric characteristics of PVA/DL-alanine polycrystals in d33 mode." *Iscience* 26.1 (2023): 105768.

5. Why isn't there a value calculated from AFM? Why wasn't lateral AFM conducted? I suggest this can add to the value of the results.

A: We thank the reviewer for the thoughtful suggestion. We did not systematically study the piezoelectricity of single MF because we have focused more on the macroscopic piezoelectricity of whole DL-alanine network, which determines the performance of piezoelectric device. To assess the piezo coefficient of single microfiber, we performed optimized vertical PFM test under different driving voltages on a single microfiber by utilizing a Bruker's Dimension Icon AFM. As presented in Figure R16, the DL alanine microfiber exhibited strong vertical responses. Uniformly strong phase responses in vertical direction (Figure R16a) indicated high out-of-plane polarizations whereas piezoelectric coefficients d_{33}^{eff} can be assessed by amplitude responses (Figure R16b) under different driving voltages. Calibrated by standard LiNbO₃ sample, this DL alanine revealed a longitudinal piezocoefficient d_{33}^{eff} of ~5.5 pm (Figure R16c).

Figure R16 (Figure S18). a. Vertical PFM phase response of single DL-alanine MF. b. Vertical PFM amplitude response of single DL-alanine MF. c. PFM amplitude responses of single DL-alanine under different driving voltages. d. Lateral PFM phase response of single DL-alanine MF. e. Lateral PFM amplitude response of single DL-alanine MF.

The lateral PFM (Figure R16d and e) was also performed on the same MF. The DL-alanine MF showed a strong lateral piezo-response as well. This excellent in-plane polarization is attributed to the growth of MF along its polar axis. As lateral PFM test usually couple both shear and transverse responses due to the complex torsional deflection of the AFM tip, it is challenging to assess specific transverse piezo-coefficient (e.g., d_{31}) and shear coefficient (e.g., d_{15}).

We added the value in the main text and highlighted it. (Page 12, *The piezoelectric coefficient, d_{33}^{eff} of single DL-alanine MFs, can be assessed by amplitude responses under different driving voltages. Calibrated by standard LiNbO_3 sample, this DL alanine MF revealed a d_{33}^{eff} of ~5.5 pm.*) Figure R16 was added in the Supplementary Information as Figure S18, together with the discussions on piezo-coefficient. (Supplementary Information, Page 21: *“Uniformly strong phase responses in vertical direction indicated high out-of-plane polarizations whereas piezoelectric coefficients d_{33}^{eff} can be assessed by amplitude responses under different driving voltages. As lateral PFM test usually couple both shear and transverse responses due to the complex torsional deflection of the AFM tip, it is challenging to assess specific transverse piezo-coefficient (e.g., d_{31}) and shear coefficient (e.g., d_{15}).”*).

6. How does the density of fibers affect the resulting current?

A: We thank the reviewer for this question. It is difficult to conclude the density effect on the current output. Since current I is defined as $I = Q/t$ and t is the response time determined by the mechanical stimulation, the current is largely determined by piezoelectric charges. On the one

hand, increasing the density of microfibers can increase the number of microfibers that contribute charge. On the other hand, higher density of fibers may reduce the strains distributed in each single microfiber under the same mechanical deformation. As a result, the piezoelectric charge contributed by each microfiber is lowered. The change of total amounts of piezoelectric charge is uncertain. At the current stage, we are not able to directly study the density effect by selectively controlling the microfiber density without affecting the microfiber orientation, size and geometry. As shown in Figure R17, changing density by varying DL-alanine concentration would significantly influence the morphology and orientation of DL-alanine MFs. It is therefore challenging to rule out other factors and directly study the density effects on piezoelectric output. We will continually explore the controllable growth of DL-alanine networks for the future work. We added the relevant discussion in the manuscript (Page 14: *“The density of DL alanine MF may affect the resulting current. Nevertheless, the density-dependent piezoelectric output would be complicated. On the one hand, higher density of DL-alanine MFs increased the number of MFs that contribute charge and improve the current output. On the other hand, higher density may also reduce the strains distributed in each single MF, and therefore lower piezoelectric charges and the current output of whole network.”*).

Figure R17. DL-alanine network with different densities of MFs.

7. I am very curious regarding the piezoelectric performance at highly stretched states. Is the AC mechanical loading concentrated in different locations than the steady state stretch? I believe that on top of the mechanical FE simulations, which are very helpful in asserting the authors' understanding of the mechanical behavior, piezoelectric simulations can be carried out and greatly improve the complete understanding of the device.

A: We thank the reviewer for the comments. During our tests and applications, the mechanical loading applied to the DL-alanine network is not alternating (AC). While we only have unidirectional tensile straining, the AC mechanical loading should involve both tensile and compressive straining. Specifically, increasing the tensile stress from zero to a certain value would stretch the piezoelectric network, while decreasing it to zero would release the network and return it to original state. Although dynamic, the strain is supposed to be concentrated on the same locations due to unidirectional loading. To convince the reviewer, we simulated strain distribution of a simplified piezoelectric network under unidirectional dynamic loading (deformation $Y \propto \sin^2(\pi t)$) and presented the results of an “X” junction at different time frames as Figure R18a. As presented, the mechanical strains concentrate in the same location during the dynamic process.

As for the piezoelectric performance, it is directly correlated to strains. Higher strain leads to higher stress (materials with same modulus), and thus larger piezoelectric potential. We then performed piezoelectric simulation of a simplified DL-alanine network consisting of regular quadrilateral units on PDMS by the commercial FEA software ANSYS 19.0 Version. A longitudinal 20% tensile strain is applied to the DL-alanine/PDMS film. The piezoelectric potential of one complete quadrilateral unit and an “X” junction are presented in Figure R18b. The majority of the unit has a low piezoelectric potential around 0.2-0.4 V, whereas the high piezoelectric response over 1 V was found near the “X” junction area, which is consistent with the simulated strain distribution in Figure R18a.

Figure R18 (Figure S17). a. Dynamic strain distribution of an “X” junction at different time frames with the network under unidirectional dynamic loading (deformation $Y \propto \sin^2(\pi t)$). b. The piezoelectric potential distribution of one complete quadrilateral unit and an “X” junction under 20% longitudinal tensile strain.

We added the dynamic mechanical simulation and piezoelectric potential simulation as Figure S17 in the supporting information. The relevant discussions are added in the main text of the manuscript with highlights. (Page 11: “Simulation of dynamic strain distribution revealed that the strains always concentrate on the same location of “X” junction during the unidirectional tensile straining (Figure S17a). As piezoelectric responses are directly correlated to strains, the simulation of piezoelectric potential demonstrated that the highest piezoelectric response is near the “X” junction area while the majority of the network has a low piezoelectric potential, which is consistent with the simulated strain distribution (Figure S17b).”).

8. Similar to point 1 - how does the flexible energy harvester compare in efficiency to others reported in literature?

A: We thank the reviewer for the question. As displayed in Table 1, most conformal/stretchable piezoelectric devices are more advantageous as sensors instead of energy harvesters, due to the limited outputs. The energy conversion efficiency is less provided in most akin work. Here, in our work, the energy conversion efficiency η of DL-alanine network/PDMS film can be estimated by comparing the input energy and output energy:

$$\eta = \frac{E_{output}}{E_{input}}$$

The energy input by tapping force (3 N) into the DL-alanine/PDMS film is equal to the energy built by strain in the network/PDMS film and can be calculated using the following equation:

$$E_{input} = \frac{1}{2}SE\varepsilon^2$$

where S is the effective volume of the film (10 $\mu\text{m} \times 1 \text{ cm} \times 1 \text{ cm}$), E is the Young's modulus (3.8 MPa measured in the manuscript), and ε is the average strain ($\sim 0.5\%$ under 3 N normal tapping force, estimated based on the film deformation). Meanwhile, the output energy of DL-alanine/PDMS thin film can be calculated by the formula:

$$E_{output} = \int_{t_1}^{t_2} \frac{V^2}{R} dt$$

R is the inner impedance of voltage meter (10 M Ω), V is the measured voltage. The E_{input} is obtained as $\sim 48 \text{ nJ}$ whereas E_{output} is 0.52 nJ. Therefore, the energy efficiency therefore is approximately 1.08%. This value is comparable to the energy efficiency (1.76%) of previously reported conformal piezoelectric devices. We added the discussion in the main text and calculation in the Supplementary Information as Supplementary Methods. (Page 16, "*The energy conversion efficiency η of DL-alanine network/PDMS film can be estimated by comparing the input energy and output energy. An estimated efficiency of approximately 1.08% is comparable to the energy efficiency of previously reported conformal piezoelectric PZT nanoribbon device (1.76%)⁴¹*".).

9. Why was a single-electrode configuration used? is it less efficient than a "regular" two electrode configuration? Also here, the schematic should be included in the main text.

A: We thank the reviewer for the question. The reviewer raised a very good point in electrode selections. The single electrode is the optimal configuration in our device design. We agree that many piezoelectric devices take a top-bottom two-electrode configuration. However, for our DL alanine structure, using single electrode is necessary for device stability and reliability. A continuous percolated Ag nanowires (NWs) film was used as the electrode to realize a fully stretchable device. Due to the open-mesh structure of the DL alanine network, it is not possible to introduce top and bottom electrodes without shorting them, particularly under tensile strain. This requirement may be circumvented by precision electrode position and insulation, but requires high engineering investment which is beyond the scope of this work. Indeed, there are literatures reporting piezoelectric devices having single electrode, particularly piezoelectric device-based electronic skins, and the advantages of single electrode configuration include higher flexibility and sensitivity. [1,2]

We added the explanation of the selection of single electrode configuration in the main text and highlight it with yellow color (Page 15, "*The single electrode configuration is desired to realize a fully stretchable device, in order to minimize the strain-related conductivity change in other two-electrode configurations.*").

References

- [1] Wang, Xiaoxiong, et al. "Bionic single-electrode electronic skin unit based on piezoelectric nanogenerator." *ACS Nano* 12.8 (2018): 8588-8596.
- [2] Liu, Qi, et al. "Wireless single-electrode self-powered piezoelectric sensor for monitoring." *ACS applied materials & interfaces* 12.7 (2020): 8288-8295.

I look forward to seeing the revised manuscript.

REVIEWER COMMENTS

Reviewer #1 (Remarks to the Author):

The authors have addressed the comments from the previous revision and performed additional characterization to improve the manuscript. The revised manuscript and new results address well most of my concerns. Although it is improved and I remain supportive, there is still a concern regarding the practical application of the piezoelectric DL-alanine microfibers network, which diminishes the broad interest and significance of the work.

I would recommend that the author consider either of the following suggestions.

1) I agree with the authors that it is challenging to fabricate a fully biodegradable stretchable tactile sensor due to the challenges in fabricating the electrodes and the limited options for choosing a biodegradable elastomer substrate. However, to make the work more significant and potentially could be used for transient bioelectronics applications, it is necessary to demonstrate, at least, the ability to grow the DL-alanine MFs on a biodegradable elastomer surface. In fact, there are several established biodegradable elastomers, such as Poly-(Glycerol Sebacate) (PGS) (1) or Poly-(Glycerol Dodecanedioate) (2), that authors can acquire and use to demonstrate the stretchability of the DL-alanine network on these substrates.

(1) Wang, Y., Ameer, G. A., Sheppard, B. J., & Langer, R. (2002). A tough biodegradable elastomer. *Nature biotechnology*, 20(6), 602-606.

(2) Ramaraju, H., Ul-Haque, A., Verga, A. S., Bocks, M. L., & Hollister, S. J. (2020). Modulating nonlinear elastic behavior of biodegradable shape memory elastomer and small intestinal submucosa (SIS) composites for soft tissue repair. *Journal of the mechanical behavior of biomedical materials*, 110, 103965.

2) If the first option is not feasible, I recommend that the authors investigate and characterize the wearable tactile device application more. Several parameters need to be analyzed, for example:

+ How long can the device produce consistent voltage outputs under the same strain? It is recommended that the authors perform a durability test of their device to assess the performance of the device over long periods.

+ It would be more significant if there is a performance comparison between DL-alanine and other piezoelectric materials (e.g., PVDF) or a commercially available tactile sensor. As seen in video 3, the

connection wires moved when the fingers were bent, which could create noise. As such, a control device would be beneficial in this case to validate the measured signal.

+ The authors use a single electrode for the device, which brings up a concern relating to external electrical noise (e.g., EMF, triboelectric) that would interfere with the output signals when the device is used for pressure sensing.

Reviewer #2 (Remarks to the Author):

The revised manuscript addressed all the issues that this reviewer raised. Therefore, I think it is ready to publish.

Reviewer #3 (Remarks to the Author):

I have received the responses and revised manuscript from the authors. I thank the authors for taking the reviews seriously and their point-by-point answer.

I have several remarks still to be considered before publication:

1. The authors should show their LiNbO₃ calibration.
2. There is a gap between the calculated d tensor by Guerin (2019) and the direction of fibers presented here. The authors claim that the vertical PFM signal is the d₃₃ coefficient, however the assigning of directions 1-2-3 to actual fiber geometry is not explicitly written anywhere (or at least I've missed it).
3. The PFM signal shown (R16/S18) clearly demonstrate that there is a bias signal from the surface (what is that surface?). Is that signal also linear with Vac? The more reasonable estimate of the amplitude is the difference between the fiber and the surface (although even this is not a complete reduction of non-piezoelectric contributions) to allow for a careful calculation of effective d coefficient (and perhaps assign it to a tensor element).
4. In the 3-point bending current experiment, is it really d₃₁ coefficient? the active mechanical deformation is stretching in the long axis therefore it is probably d₃₃.

Reviewer #1 (Remarks to the Author):

The authors have addressed the comments from the previous revision and performed additional characterization to improve the manuscript. The revised manuscript and new results address well most of my concerns. Although it is improved and I remain supportive, there is still a concern regarding the practical application of the piezoelectric DL-alanine microfibers network, which diminishes the broad interest and significance of the work.

I would recommend that the author consider either of the following suggestions.

A: We express our gratitude to the reviewer for thoroughly examining our point-by-point responses. We also appreciate the additional suggestions from the reviewer to further improve the broad interest and significance of this work.

1) I agree with the authors that it is challenging to fabricate a fully biodegradable stretchable tactile sensor due to the challenges in fabricating the electrodes and the limited options for choosing a biodegradable elastomer substrate. However, to make the work more significant and potentially could be used for transient bioelectronics applications, it is necessary to demonstrate, at least, the ability to grow the DL-alanine MFs on a biodegradable elastomer surface. In fact, there are several established biodegradable elastomers, such as Poly-(Glycerol Sebacate) (PGS) (1) or Poly-(Glycerol Dodecanedioate) (2), that authors can acquire and use to demonstrate the stretchability of the DL-alanine network on these substrates.

(1) Wang, Y., Ameer, G. A., Sheppard, B. J., & Langer, R. (2002). A tough biodegradable elastomer. *Nature biotechnology*, 20(6), 602-606.

(2) Ramaraju, H., Ul-Haque, A., Verga, A. S., Bocks, M. L., & Hollister, S. J. (2020). Modulating nonlinear elastic behavior of biodegradable shape memory elastomer and small intestinal submucosa (SIS) composites for soft tissue repair. *Journal of the mechanical behavior of biomedical materials*, 110, 103965.

A: We thank the reviewer for this thoughtful suggestion to improve the significance of this work. As we agreed in our previous responses, the importance of demonstrating transient bioelectronics application is indisputable. We thoroughly read through the references suggested by the reviewer, and highly appreciate these biodegradable elastomers Poly-(Glycerol Sebacate) (PGS) in reference 1 and Poly-(Glycerol Dodecanedioate) (PGD) in reference 2 as potential substrates. Syntheses of these elastomers are practical and smart for soft materials labs. However, there are currently critical technical challenges in using these two materials for consistent network growth, such as making sub ten nanometer flatness on the surface and suppressing large swelling in ethanol-water biphasic solution. Using these new degradable substrate materials demands amounts of additional efforts, and is not aligned with the focus of our current work. Therefore, we cited these two valuable references in the main text and indicated the future promise of

transient bioelectronic applications enabled by these biodegradable elastomers. (Page 17-18, “*Particular efforts to engineer biodegradable elastomers such as poly-(glycerol sebacate) (PGS)⁵⁷ and poly-(glycerol dodecanedioate) (PGD)⁵⁸ as growth substrates for DL alanine network can greatly facilitate the realization of a fully biodegradable stretchable tactile sensor.*”)

2) If the first option is not feasible, I recommend that the authors investigate and characterize the wearable tactile device application more. Several parameters need to be analyzed, for example:

A: We thank the reviewer for constructive comments on the second aspect. We added more characterizations as suggested.

+ How long can the device produce consistent voltage outputs under the same strain? It is recommended that the authors perform a durability test of their device to assess the performance of the device over long periods.

A: We thank the reviewer for the suggestion of durability test. We conducted the durability test by constantly straining our DL-alanine network-based devices for 20% and bending it through a computer-controlled actuator (LinMot USA, Inc) at a frequency of 1 Hz. The voltage outputs of the device over 1500 continuous cycles were collected and presented as Figure R1 below. No significant reduction in the outputs of our device is witnessed, as illustrated in Figure R1a. Further analyses in detail showed that the outputs in the first 20 cycles (Figure R1b) match well with the counterparts in the last 20 cycles (Figure R1c). As a result, our durability test validated the stable performance of our stretchable DL alanine network and backed its potential practical applications. We added Figure R1 as Figure S25 in the Supplementary Information and added the corresponding discussion in the main text. (Page 16, “*After being strained 20%, this piezoelectric tactile sensor based on DL-alanine network was still able to generate consistent voltage signals with no significant reduction in amplitude over 1500 cyclic bending, revealing excellent durability (Figure S25).*”).

Figure R1 (Figure S25). Long-term durability test of DL alanine network based piezoelectric device. **a.** Voltage outputs of 20% strained device under 1500 cyclic bending. **b.** Voltage outputs under first 20 cyclic bending. **c.** Voltage outputs under last 20 cyclic bending.

+ It would be more significant if there is a performance comparison between DL-alanine and other piezoelectric materials (e.g., PVDF) or a commercially available tactile sensor. As seen in video 3, the connection wires moved when the fingers were bent, which could create noise. As such, a control device would be beneficial in this case to validate the measured signal.

A: We thank the reviewer for the comment. To improve the reliability of our device measurement, we used both commercial PVDF thin film (8 μm , PolyK Technologies, LLC) and bare PDMS film as control devices for comparison and noise exclusion. Same as the DL alanine network, both PVDF film and bare PDMS film were integrated with the Ag NW electrode to make single electrode control devices. The devices were attached to the knuckles under repeated bending and the results were presented as Figure R2 below. As shown in Figure R2a, a peak-to-peak voltage outputs of ~ 1 V were generated by the PVDF-based device. The higher voltage output from the PVDF device was due to the larger thickness and higher piezoelectric coefficient of the PVDF film. This output change validated that the measured signals were from piezoelectric components instead of noises due to wire movement. Otherwise, similar outputs of PVDF and DL alanine network devices would be received. Furthermore, the bare PDMS based single electrode device, although stretchable and conformal, only generated noise-level signals under knuckle

bending, indicating the negligible contributions from noises or other artifacts due to wire movement or other extrinsic issues in voltage outputs measurement. We added Figure R2 as Figure S26 in the Supplementary Information and added the corresponding discussion in the main text. (Page 16, “Control devices made of commercial PVDF and bare PDMS thin films (Figure S26) validated the normal function of single electrode configuration and excluded the noise contributions from wire movement or other measurement-related artifacts.”)

Figure R2 (Figure S26). **a.** Schematic of PVDF based single electrode piezoelectric device and its output under knuckle bending. **b.** Schematic of bare PDMS based single electrode piezoelectric device and its output under knuckle bending.

+ The authors use a single electrode for the device, which brings up a concern relating to external electrical noise (e.g., EMF, triboelectric) that would interfere with the output signals when the device is used for pressure sensing.

A: We thank the reviewer for the comment. As pointed out in our previous responses, using of signal electrode configuration is advantageous for this stretchable system, because it is not feasible to introduce top and bottom percolated Ag NWs electrodes (percolated Ag NW electrode is necessary for a fully stretchable device) without shorting them due to the open-mesh structure of the DL alanine network, particularly under tensile strain.

In fact, external electrical noises are unavoidable for all electronic systems regardless of electrode configurations. They can be minimized to make negligible interference to the real signals. The two types of electrical noises the reviewer mentioned, namely, electromagnetic field noise and triboelectric noise, have been carefully minimized

in all of our device tests. For electromagnetic field noise, which usually have high frequency or unmatched frequency with the targeted output signals, they can be filtered and reduced by the measurement system. Here, low noise mode, low input offset current and low-pass filter in the pre-amplifier are all adapted in our tests to remove these noises from the measurement signal, as we have emphasized in the main text of our manuscript (Page 20, “To maximize the accuracy of measurement, low noise mode, low input offset current and low-pass filter in the pre-amplifier are adapted to remove noise from the measurement signal.”). The generation of triboelectric noises requires the contact and separation of two dissimilar dielectric materials, such as device and skin. Thanks to the stretchability, conformality and tissue-comparable modulus, our DL alanine network piezoelectric device can be conformally fixed to knuckle, wrist and skins even under large strains (as demonstrated in Figure 4 and Figure S27), which well prevented the significant triboelectric noises. This is also validated in Figure R2, where bending a bare PDMS single-electrode device by knuckle generated negligible outputs. As a result, external electrical signals due to electromagnetic field and triboelectricity can be minimized and will not significantly interfere with the output signals in our DL alanine network devices. We added relevant discussion in the main text. (Page 16-17, “*In our measurements, potential interference from external signals due to electromagnetic field or triboelectricity were minimized by pre-amplifier filtering or minimizing contact-separation electrification, respectively, in the measurement of piezoelectric tactile sensor signals.*”)

Reviewer #2 (Remarks to the Author):

The revised manuscript addressed all the issues that this reviewer raised. Therefore, I think it is ready to publish.

A: We highly thank the reviewer for reading through our point-by-point responses and the recommendation to publication.

Reviewer #3 (Remarks to the Author):

I have received the responses and revised manuscript from the authors. I thank the authors for taking the reviews seriously and their point-by-point answer.

A: We thank the reviewer for carefully reading through our point-by-point responses and even references. We highly appreciate this scientific rigor and the additional efforts/comments to further improve our manuscript quality.

I have several remarks still to be considered before publication:

1. The authors should show their LiNbO3 calibration.

A: We thank the reviewer for the comment. We utilized a periodically poled lithium niobate (PPLN) specimen from Bruker as a reference sample for calibration (Figure R3). This PPLN has a known effective piezoelectric coefficient $d_{33} = 7.5$ pm/V. First, we measured the vertical piezoelectric response of the calibration sample (Figure R3a and b). We then estimated the slope k_{PPLN} of the linear fitting of the piezoresponse amplitude versus the applied voltage (Figure R3c). Afterward, the calibration factor α can be obtained as:

$$\alpha = k_{PPLN} / d_{33\text{-reference}}$$

in which $d_{33\text{-reference}}$ is the known piezoelectric coefficient of the reference sample. Therefore, the d_{33} of experimental sample is given by the consideration of calibration factor α :

$$d_{33} = \frac{d_{\text{measured}}}{\alpha \times V_{AC}}$$

in which d_{measured} is the and V_{AC} is the alternating driving voltage. As shown in Figure R3c, our calculated piezocoefficient of PPLN ($k_{PPLN} = 8.2$ pm/V) is very close to the reference value ($d_{33\text{-reference}} = 7.5$ pm/V). As a result, we only have a calibration factor $\alpha = 8.2/7.5 \approx 1.1$. This calibration factor has been included in the calculation of effective d_{33} of DL alanine MFs. We added Figure R3 as Figure S19 in Supplementary Information and the calibration process in Supplementary Methods in Supplementary Information (Supplementary Information, Page 3).

Figure R3 (Figure S19). PFM calibration by standard periodically poled LiNbO3 (PPLN) sample. **a.** Vertical amplitude response of PPLN sample under 1 V driving voltage. **b.** Vertical phase response of PPLN sample under 1 V driving voltage. **c.** Linear fitting of the piezoresponse amplitude versus the applied voltage.

2. There is a gap between the calculated d tensor by Guerin (2019) and the direction of fibers presented here. The authors claim that the vertical PFM signal is the d_{33} coefficient,

however the assigning of directions 1-2-3 to actual fiber geometry is not explicitly written anywhere (or at least I've missed it).

A: We thank the reviewer for examining references. Indeed, our measured effective d_{33} value by PFM (~ 5.5 pm/V) is not exactly the same as the predicted d_{33} of DL alanine in the reference (10.4 pm/V) [1]. While the predicted d_{33} based on DFT calculation can be easily influenced by system size (e.g., number of unit cells) and function approximations, how many unit cells used in the DFT calculation are not provided in the cited paper by Guerin et al [1]. It is reasonable to see a difference between measured results on bulk samples and computed results based on limited unit cells. Besides, the quantitative measurement by vertical PFM is subjected to tip-sample interaction, tip geometry, sample geometry, e.g. In fact, the effective d_{33} measurement through PFM by Guerin (2019) [1] revealed a wide range (6.8-12.4 pm/V, Figure S6 in the Supplementary Information of reference [1]). Therefore, it is common to see that the calculated value and measured value are different due to the acceptable errors in calculations and experiments.

The vertical PFM measures effective longitudinal piezoelectric coefficients. Since DL-alanine crystals have a point group of $mm2$ (orthorhombic system), which has non-zero d_{33} , the out of plane direction is thus defined as the effective “3” direction. Effective directions in relation to the DL alanine MF are defined and presented in the figure (Figure R4) below. This definition of effective directions of DL alanine MF is consistent with the effective directions in Figure 3d.

Figure R4. Vertical amplitude response of individual DL alanine MF. Effective directions in relation to the DL alanine MF are defined by the white arrows. The out of plane direction is thus defined as effective “3” direction.

We added the discussions on difference between calculated value in reference and our measured effective piezo-coefficient in the main text and also defined the effective directions in relation to the DL alanine MFs. Figure R4 is also inserted as Figure S13d in the Supplementary Information (Page 12, “Since DL-alanine crystals has non-zero d_{33} ... the out of plane direction is thus defined as the effective “3” direction together with quantified piezoelectric coefficient as effective d_{33} ...” “The measured d_{33} was reasonably smaller than the calculated value of 10.34 pm/V from a perfect structure³⁹, given the

practical measurement conditions in PFM with tip-sample interactions and structural imperfection.”).

References:

[1] Guerin, Sarah, et al. "Racemic amino acid piezoelectric transducer." *Physical review letters* 122.4 (2019): 047701

3. The PFM signal shown (R16/S18) clearly demonstrate that there is a bias signal from the surface (what is that surface?). Is that signal also linear with V_{ac} ? The more reasonable estimate of the amplitude is the difference between the fiber and the surface (although even this is not a complete reduction of non-piezoelectric contributions) to allow for a careful calculation of effective d coefficient (and perhaps assign it to a tensor element.

A: We thank the reviewer for reading through the details. We utilized the single-side polished highly doped ($0.001\text{--}0.005\ \Omega\cdot\text{cm}$) silicon substrate (University Wafer). Highly doped silicon substrate has been commonly used in PFM measurement. Due to its flat surface, we were able to directly grow DL alanine network on the doped Si substrate and make PFM measurement on the same substrate afterwards. This substrate signal is not linear with driving voltages. To convince the reviewer, we performed vertical PFM on bare doped Si substrate under different driving voltages. The results are presented as Figure R5. While the applied voltage is boosted from from 1 V to 5 V, the substrate signal remained relatively stable around 13 pm/V, with only a slight increase under 5 V driving voltage ($\sim 16\ \text{pm/V}$).

Figure R5. vertical PFM amplitude response of bare doped Si substrate under different driving voltages.

We indeed subtracted the substrate signal when assessing the effective d_{33} coefficient. In Figure S18c, the plotting of vertical response under different driving voltage is after subtracting the background contribution. The first point in the curve corresponds to $5.2\pm 0.4\ \text{pm/V}$ under driving voltage of 1V, which is obtained by the measured value

(Figure S18b) (17.5 ± 0.4 pm/V) subtracting the substrate background contribution (12.3 pm/V). To avoid misunderstanding, we revised Figure S18b, highlighted the subtraction of substrate contribution, and presented as Figure R6 below. Not only subtracting the substrate signal, the calibration factor obtained by LiNbO_3 calibration was also applied (Response in question 1). Therefore, the measurement of effective d_{33} coefficient is reliable.

Figure R6 (Revised Figure S18b). Vertical amplitude response of individual DL alanine MF. The subtraction of substrate background contribution in piezoresponses is highlighted.

Because of the shape and geometry of DL alanine MFs, it might be challenging to directly assign this effective d_{33} coefficient to a specific tensor element in the coefficient matrix. Unlike 2D materials and thin films where a specific piezo-coefficient can be dominant during PFM characterization given the flat surface and simple geometry, this is more complex when it comes to our DL alanine MF with a trapezoid cross-section. In addition to the longitudinal components, shear components and transverse components may also contribute to the vertical PFM response when the tip crosses DL alanine MF. We would thus remain the claim as effective d_{33} instead of a specific tensor element. We added the discussion to the main text. (Page 12, “*Since DL-alanine crystals has non-zero d_{33} and the transverse and shear piezoelectric components in addition to the longitudinal component may contribute to the vertical PFM response due to the shape and geometry of DL alanine MFs, the out of plane direction is thus defined as the effective “3” direction together with quantified piezoelectric coefficient as effective d_{33} rather than assigned to a specific tensor element in the coefficient matrix.*”)

4. In the 3-point bending current experiment, is it really d_{31} coefficient? the active mechanical deformation is stretching in the long axis therefore it is probably d_{33} .

A: We thank the reviewer for the comment. While what we measured is effective d_{13} of the whole DL alanine network based on the definition of piezoelectric coefficient, the reviewer is also correct that the longitudinal polarization of the whole film certainly contributed to the measured piezoelectric response due to the in-plane stretching of the

network under tapping force. In the current measurement experiment (Figure 3d), according to piezoelectricity definition ($P_i = d_{ij} \cdot \sigma_j$), a normal tapping force applied along “3” direction indicates a stress σ_3 ($j = 3$). As two parallel electrodes were applied perpendicularly to the effective direction “1”, the polarization that we collected is P_1 ($i = 1$). Since i and j are known as 1 and 3 here, respectively, the measured effective coefficient is transverse coefficient d_{13} . Nevertheless, given our experiment setup and film configuration, tapping the DL network/PDMS film with two fixed ends can also lead to the in-plane stretching of the film along the effective “1” direction, and therefore, longitudinal piezoelectric component (d_{11}) would also contribute to the measured piezoelectric response. We added this discussion in the main text (Page 14, “*While the measured effective piezoelectric coefficient is the transverse d_{13} of the whole DL alanine network based on the definition, longitudinal component (effective d_{11}) of the whole film may also have small contributions to the measured piezoelectric response due to the in-plane stretching of the network under tapping force.*”).

REVIEWERS' COMMENTS

Reviewer #1 (Remarks to the Author):

the authors have addressed all concerns and the paper is good for publication now

Reviewer #3 (Remarks to the Author):

The authors have addressed all my concerns